# The L27 domain of MPP7 enhances TAZ-YY1 cooperation to renew muscle stem cells

Anwen Shao[1], Joseph L Kissil[2] & Chen-Ming Fan [ID] [1,3][✉]

## Abstract

**Stem cells regenerate differentiated cells to maintain and repair tissues and organs. They also replenish themselves, i.e. self-renew, to support a lifetime of regenerative capacity. Here we study the renewal of skeletal muscle stem cell (MuSC) during regeneration. The transcriptional co-factors TAZ/YAP (via the TEAD transcription factors) regulate cell cycle and growth while the transcription factor YY1 regulates metabolic programs for MuSC activation. We show that MPP7 and AMOT join TAZ and YY1 to regulate a selected number of common genes that harbor TEAD and YY1 binding sites. Among these common genes, *Carm1* can direct MuSC renewal. We demonstrate that the L27 domain of MPP7 enhances the interaction as well as the transcriptional activity of TAZ and YY1, while AMOT acts as an intermediate to bridge them together. Furthermore, MPP7, TAZ and YY1 co-occupy the promoters of *Carm1* and other common downstream genes. Our results define a renewal program comprised of two progenitor transcriptional programs, in which selected key genes are regulated by protein-protein interactions, dependent on promoter context.**

**Keywords** Stem Cells; Renewal Division; Muscle Regeneration; Tiered Regulation; Transcription
**Subject Categories** Chromatin, Transcription & Genomics; Signal Transduction; Stem Cells & Regenerative Medicine

## Introduction

Stem cells are critical for tissue homeostasis. Depending on the tissue, resident stem cells cycle constantly, periodically, or rarely under physiological conditions (Fuchs and Blau, 2020). Upon injury, stem cells can enter a faster or longer proliferative cycle to produce differentiated cells for repair. Stem cells also replenish themselves, i.e. renewal, to sustain a lifetime of tissue homeostasis and regeneration. Here we focus on MuSCs, which are important to muscle homeostasis and regeneration, as well as in muscle diseases, cancers, and aging (Relaix et al, 2021; Sousa-Victor et al, 2022).

The main source of MuSCs are PAX7-expressing (PAX7[+]) cells known as satellite cells (Mauro, 1961), as elucidated by lineage tracing and cell ablation studies in mice (Lepper et al, 2011; Murphy et al, 2011; Sambasivan et al, 2011). They are attached to the muscle fiber via the apical adherens junction (AJ) and situated on the basal extracellular matrix (ECM) surrounding the myofiber. Loss of M- and N-cadherins (AJ components) leads to MuSC activation and incorporation into myofiber, as well as supernumeral MuSCs (Goel et al, 2017). Loss of the ECM-receptor β1-integrin leads to minimal incorporation of MuSC into myofibers and loss of MuSCs (Rozo et al, 2016). Cadherins and integrins are tethered to filamentous actins (F-actins) and actin dynamics affect the state/fate of MuSCs. For example, Rac and Rho, two small GTPases regulating actin polymerization at cell projection and cortex, have been implicated in MuSC quiescence and activation, respectively (Kann et al, 2022). The F-actin-tethered mechanosensitive $Ca^{2+}$ channel Piezo1, which regulates Rho and MuSC cell projections, also facilitates MuSC activation (Hirano et al, 2023; Ma et al, 2022).

Non-canonical Wnt4 signaling has been implicated in suppressing the mechano-responsive yes-associated transcriptional co-factor, YAP (Eliazer et al, 2019). YAP and related TAZ (also known as WWTR1; collectively, YAP/TAZ) are co-activators for the TEAD family (TEAD1-4) of transcription factors involved in driving cell growth and proliferation (Ma et al, 2019; Pan, 2022). Indeed, YAP overexpression promotes myoblast proliferation (Judson et al, 2012; Tremblay et al, 2014) by activating cell growth/cycle genes (Sun et al, 2017). Conditional inactivation of *Yap* in MuSCs (*Yap* cKO) compromises muscle regeneration, but *Taz* germline mutants do not display regeneration defects (Sun et al, 2017). Although Rho is a known regulator of YAP's transcriptional activity, the mechanisms underlying YAP/TAZ's mechano-sensitivity is incompletely understood (Panciera et al, 2017) and yet to be explored in MuSCs.

A conserved pathway that restrains YAP/TAZ activity is the Hippo kinase cascade (Ma et al, 2019; Pan, 2022). When the Hippo pathway is activated, typically via cell-junction machineries, its most distal kinases LATS1/2 phosphorylate YAP/TAZ to promote their cytoplasmic retention and degradation, thereby precluding their nuclear functions. In mammals, Angiomotin (AMOT) family members (AMOT, AMOTL1, and AMOTL2) interact with multiple Hippo components at cell junctions and on F-actins (Moleirinho et al, 2014). AMOT's LPTY and PPxY motifs bind to the WW domain of YAP/TAZ. AMOT can also bind to F-actins unless phosphorylated by LATS1/2. Thus, AMOTs can be considered a

[1]Department of Embryology, Carnegie Institution for Science, 3520 San Martin Drive, Baltimore, MD 21218, USA. [2]Department of Molecular Oncology, The H. Lee Moffitt Cancer Center, 12902 USF Magnolia Drive, Tampa, FL 33612, USA. [3]Department of Biology, Johns Hopkins University, 3400 N Charles Street, Baltimore, MD 21218, USA.
[✉]E-mail: fan@carnegiescience.edu

mechanical interpreter which retains YAP/TAZ at cell junction and on/off of F-actin in the cytoplasm. However, the non-phosphorylated AMOT has also been shown to increase nuclear YAP in certain cell types (Moleirinho et al, 2017; Yi et al, 2013). In this scenario, whether AMOT also modulates YAP/TAZ's transcriptional activity and target gene selectivity is unknown.

Prior to establishment of the connection to the Hippo pathway, AMOT was characterized as an angiostatin-binding protein that helps localize Rho to the leading edge of migrating endothelial cells (Moleirinho et al, 2017). AMOT also binds and inhibits Rich1 (a GTPase activating protein)-mediated hydrolysis of Rac1 to regulate tight junctions (TJs). A lesser studied aspect of AMOT is its association with membrane palmitoylated protein 7 (MPP7) (Wells et al, 2006). MPP7 has been implicated in maintaining TJ and AJ via binding to MPP5/Crumb (Stucke et al, 2007) and DLG/LIN7 (Bohl et al, 2007), respectively, but these roles have not been linked to AMOT. Based on protein interaction and RNAi data, we had previously proposed that MPP7 and AMOT act together with YAP in the nucleus to support MuSC proliferation and renewal (Li and Fan, 2017). The mechanism underlying their coordinated regulation of MuSC is largely unknown.

While YAP has been mostly characterized as a transcriptional co-activator, it can also function as a co-repressor for cell cycle inhibitor genes in human Schwann cells (Hoxha et al, 2020). There, YAP co-occupies genomic regions with the transcription factor Ying-Yang 1 (YY1) and the enhancer of zeste homolog 2 (EZH2) for gene repression. In MuSCs, *Yy1* represses mitochondrial genes while activating glycolytic genes involved in metabolic reprogramming during MuSC activation (Chen et al, 2019). As YAP/TAZ governs cell cycle/growth genes and YY1 metabolic genes in MuSCs, their parallel actions should help ensure transition into an activated state. Whether their actions intersect, via transcriptional activation or repression to direct MuSC renewal has not been explored.

To summarize, MPP7, AMOT, TAZ(YAP), and YY1 have been shown to possess separate as well as overlapping functions in different processes and cell types. Here we demonstrate their convergent function in MuSC renewal. Through interrogation of RNA-seq data from *Mpp7*, *Amot*, *Taz;Yap*, and *Yy1* cKO MuSCs we identified common downstream genes that harbor both TEA-D(YAP/TAZ) and YY binding sites in their promoters. One of these, *Carm1*, plays a known role in MuSC renewal (Kawabe et al, 2012). Using a combination of assays, we demonstrate the convergence of MPP7, TAZ and YY1 to *Carm1* promoter to drive high levels of expression, whereas AMOT bridges their interactions. Importantly, the L27 domain of MPP7 not only enhances the interaction between TAZ and YY1 but also possesses intrinsic transcriptional activity. Moreover, nuclear entry of MPP7 and AMOT is regulated by the state of F-actin in MuSCs. Together, we propose a model in which MuSC renewal involves mechanosensitive regulation of AMOT and MPP7 which facilitates the cooperation between TAZ(YAP) and YY1, leading to high levels of *Carm1* expression that drives MuSC renewal.

# Results

## *Mpp7* plays a role in MuSC renewal

To determine the role of *Mpp7* in the MuSC, we generated compound mice carrying *Mpp7*$^{flox}$ and *Pax7*$^{CreERT2}$ (Lepper et al, 2009) alleles. The recombination of flanking loxP sites *in the Mpp7*$^{flox}$ allele predicts a frameshift with an early stop codon (Fig. 1A). The treatment regimen of tamoxifen (TMX)-induced *Mpp7* cKO and control (*Pax7*$^{CreERT2/+}$) mice is depicted in Fig. 1B; *Rosa-YFP* (Srinivas et al, 2001) was included as a cellular marker and to facilitate fluorescence-activated cell sorting (FACS). A knockout efficiency of ~92% was achieved based on the loss of MPP7 immunofluorescent (IF) signal in FACS-isolated *Mpp7* cKO cells (Appendix Fig. S1A,B; "Methods"). Without injury, PAX7$^+$ cell numbers in the tibialis anterior (TA) muscles were not different between control and *Mpp7* cKO 30 days (d) after TMX (Appendix Fig. S1C), indicating that *Mpp7* is not required to maintain quiescent MuSCs. To assess regeneration (Fig. 1B), additional TMX injections after injury were included to boost cKO efficiency. At 5 days post-injury (dpi), the *Mpp7* cKO had smaller regenerated myofibers and lower PAX7$^+$ cell density (Fig. 1C–F) compared to the control. At 21 dpi (Fig. 1G,H; Appendix Fig. S1D,E), regenerated myofibers remained smaller and the quiescent PAX7$^+$ cell density lower in the *Mpp7* cKO compared to those in the control. Using in vivo EdU incorporation accumulated over the first 5 d of regeneration, we found fewer proliferated YFP$^+$ cells in the *Mpp7* cKO than those in the control (Fig. 1I; Appendix Fig. S1F). The majority of renewed quiescent MuSCs are documented to be derived from cell divisions at d 5 and onwards after injury (Cutler et al, 2022). Our data therefore suggest either that the early proliferation defect (1–5 dpi) of *Mpp7* cKO leads to the reduction of renewed quiescent SCs at 21 dpi or that *Mpp7* also acts in later renewal divisions prior to quiescence, and possibly both.

FACS-isolated YFP$^+$ cells cultured in vitro also showed a smaller EdU$^+$ fraction in the *Mpp7* cKO compared to that seen in the control, but no difference in programmed cell death (PCD) was found (Appendix Fig. S1G-I). We next used the established single myofiber (SM) assay(Zammit et al, 2004) to assess the relative fractions of renewal (PAX7$^+$), progenitor (PAX7$^+$MYOD$^+$), and differentiation-committed (MYOD$^+$) cell fates derived from MuSCs after 3 days of culture (Fig. 1J; Appendix Fig. S1J). *Mpp7* cKO had fewer progenitor and renewed cells and more differentiation-committed cells, compared to the control. Thus, *Mpp7* supports MuSC proliferation and renewal during regeneration.

## MPP7's PDZ and L27 domains are critical for its function in MuSCs

As MPP7 has been documented to maintain AJs in various cell lines (Bohl et al, 2007), we examined AJs in *Mpp7* cKO MuSCs. Immediately after SM isolation, *Mpp7* cKO MuSCs displayed normal apically localized M-cadherin, N-cadherin, β-catenin, and PAR3 (Appendix Fig. S2A), indicating no gross disruption of AJs. Consistently, compromised AJs leads to supernumeral MuSCs (Goel et al, 2017), which was not found in *Mpp7* cKO (Appendix Fig. S1C).

We next investigated the requirement of different MPP7 domains for MuSC-derived fates. MPP7 is composed of an L27, a PDZ, an SH3, and a GUK (last two together as SH3GUK) domains (Fig. 2A). The L27 domain interacts with DLG/LIN7 for AJ targeting (Bohl et al, 2007), the SH3 domain interacts with MPP5/Crumb for TJ targeting (Stucke et al, 2007), and the PDZ domain has no assigned binding partner/s to date. To assess function, we devised an in vitro complementation assay utilizing transient

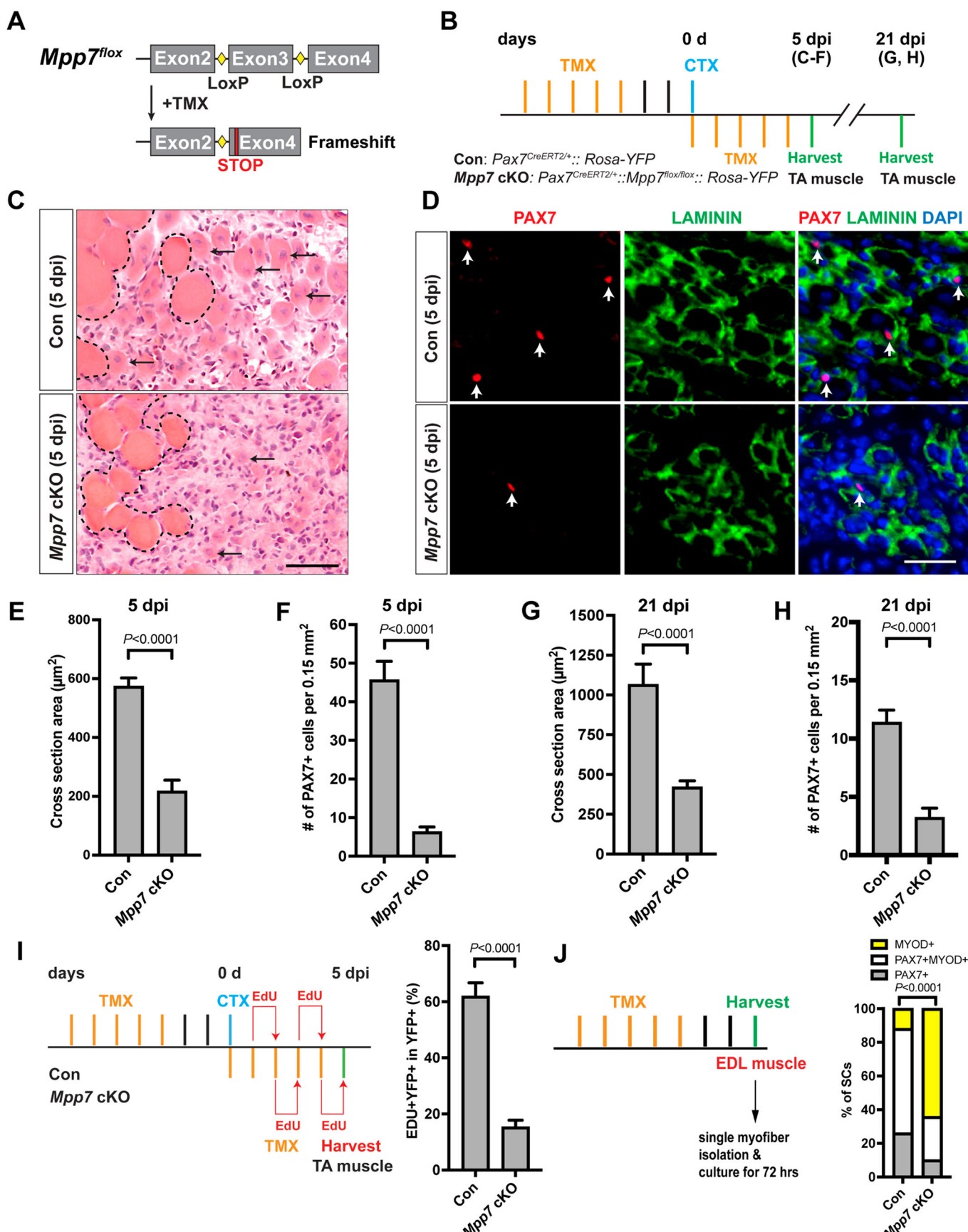

**Figure 1. Mpp7 cKO in Pax7$^+$ MuSCs shows defects in regeneration and MuSC renewal.**

(A) Diagram of *Mpp7* floxed allele (*Mpp7$^{flox}$*; loxP, yellow diamond) for tamoxifen (TMX) inducible Cre-mediated cKO. After recombination, out of-frame (Frameshift) joining of Exons 2 and 4 introduces an early stop codon (STOP). (B) Regimen of TMX administration, cardiotoxin (CTX) injury, and tibialis anterior (TA) muscle harvest; d, day; dpi, days post injury. Genotypes of control (Con; *Pax7$^{CE/+}$::Rosa-YFP*) and *Mpp7* cKO are indicated. (C–F) *Mpp7* cKO regeneration defects at 5 dpi: Representative images of H&E histology (C), immunofluorescence (IF) for PAX7 and LAMININ (D, with DAPI), and quantification of regenerated myofiber cross-sectional area (E) and of PAX7$^+$ MuSC density (F). Black arrows indicate regenerated myofibers; dashed lines, boundary of injury; white arrows, PAX7$^+$ MuSCs; (G, H) Quantifications of regenerated myofiber cross-sectional area (G) and PAX7$^+$ MuSC density (H) at 21 dpi. (I) Regimen of in vivo EdU incorporation to assess the percentage of proliferated YFP-marked cells at 5 dpi; quantification to the right. (J) Regimen of cell fate determination using SM culture. Cell fates were assessed by IF of PAX7 and MYOD; quantification to the right; keys at the top. Data information: Scale bars = 50 μm in (C) and 25 μm in (D). (C–I) N = 5 mice in each group. (J) N = 3 mice in Con group, of total 601 cells; N = 4 mice *Mpp7* cKO group, of total 517 cells. Data represent means ± SD; Student's *t* test (two-sided) in (E-I). Chi-square test in (J). Source data are available online for this figure.

transfection of full-length (WT), L27-deleted (ΔL27), PDZ-deleted (ΔPDZ), and SH3GUK-deleted (ΔΔSH3GUK) Mpp7 expression plasmids (Fig. 2A) into *Mpp7* cKO MuSCs in SM culture (Fig. 2B). All these forms of Mpp7 exhibit similar cellular distribution (Appendix Fig. S2B). WT and ΔΔSH3GUK Mpp7 rescued both progenitor and renewal fates, ΔL27 MPP7 rescued progenitor but not renewal fate, and ΔPDZ MPP7 rescued neither (Fig. 2C; Appendix Fig. S2C). Thus, MPP7's PDZ domain is critical for both renewal and progenitor fates, whereas its L27 domain is uniquely required for renewal.

## The PDZ-binding motif (PDM) of AMOT binds to the PDZ of MPP7 and is critical for its function

We have previously shown that exogenously expressed MPP7 and AMOT can co-immunoprecipitate (co-IP) with each other in 293T cells (Li and Fan, 2017). How they interact and whether *Amot* plays a role in MuSCs are unknown. By co-IP, we find that MPP7 and AMOT bind to each other via their respective PDZ and PDM domains (Fig. 2D). *Amot* cKO mice (same strategy as for *Mpp7* cKO mice) showed regeneration defects similar to those observed in the *Mpp7* cKO mice: Smaller myofibers, fewer PAX7$^+$ cells, and a reduced EdU$^+$ fraction (Fig. 2E–H). In SM culture, the *Amot* cKO cells showed reduced fractions of progenitor and renewal fates (Appendix Fig. S2D). By complementation, WT but not ΔPDM Amot could rescue these defects (Fig. 2I; Appendix Fig. S2E). When examining their protein levels we found that MPP7 was not affected in the *Amot* cKO cells (Fig. 2J) but AMOT levels were reduced in the *Mpp7* cKO cells (Fig. 2K). Thus, the interacting domains of MPP7 and AMOT share a similar role for both progenitor and renewal fates, and AMOT protein levels appear to depend on MPP7 (Appendix Fig. S2G).

## *Mpp7* cKO and *Amot* cKO MuSCs share differentially expressed genes

We next performed RNA-seq to determine differentially expressed genes (DEGs) in *Mpp7* cKO and *Amot* cKO, relative to the control (Fig. 3A–C). *Mpp7* cKO had 58 DEGs and *Amot* cKO 66 DEGs (*Padj* < 0.05 cutoff). Thirty-five of the identified DEGs intersect, with 15 of these downregulated (Fig. 3D; Dataset EV1); non-intersecting DEGs likely reflect separate roles of *Mpp7* and *Amot*. *Amot* itself is not differentially expressed in the *Mpp7* cKO cells, indicating that reduced AMOT levels in these cells likely occurs post-transcriptionally. As such, some of the shared DEGs observed in the *Mpp7* cKO cells may reflect reduced levels of AMOT. GO-

term analysis revealed enrichment for estrogen receptor (ESR) signaling, mitochondria biogenesis, and small GTPases (Fig. 3E). Genes in these GO-term have not been studied in MuSC, with the exception of *Carm1* (or *Prmt4*). CARM1 is an arginine methyl transferase that can methylate PAX7, which then recruits epigenetic regulators to activate de novo committed satellite myogenic cells (Kawabe et al, 2012). *Carm1* cKO mice also display reduced regenerative myofiber size and PAX7$^+$ MuSC number. We confirmed that CARM1 was reduced in *Mpp7* cKO and *Amot* cKO MuSCs by six- and fivefold, respectively (Fig. 3F,G; Appendix Fig. S3A).

Importantly, we found that forced expression of Carm1 was sufficient to rescue the defect of *Mpp7* cKO MuSCs using the SM transfection assay (Fig. 3H; Appendix Fig. S3B); this result alone does not exclude potential contributions by other DEGs in vivo. We next constructed a luciferase reporter driven by a putative promoter region (−630 to +15 bp) of *Carm1* (i.e., Carm1-reporter) and showed that it could be activated by WT Mpp7 (Fig. 3I), but not by Mpp5 (Appendix Fig. S3C) in 293 T cells, indicating a selectivity of this reporter for Mpp7. In addition, ΔΔSH3GUK Mpp7 activated the reporter similarly to the WT Mpp7, whereas ΔL27 Mpp7 only weakly activated and ΔPDZ Mpp7 did not activate the reporter (Appendix Fig. S3D), revealing the requirement of PDZ and L27 domains of MPP7 for Carm1-reporter activation. Thus, *Carm1* appears to function as central effector gene downstream of *Mpp7* to support MuSC-derived progenitor and renewal fates, as demonstrated by the SM complementation assay.

## The regulatory network of *Yap* and *Taz* overlaps with those of *Mpp7* and *Amot*

To understand Mpp7-mediated Carm1-reporter activation, we considered the following: (1) RNAi-mediated knockdown of *Mpp7* reduced nuclear YAP in myoblasts (Li and Fan, 2017), (2) MPP7-PDZ binds to AMOT-PDM (herein), and 3) AMOT binds to YAP and can increase nuclear YAP in certain cell types (Moleirinho et al, 2014). If the transcriptional role of MPP7 is linked to YAP/TAZ via AMOT, they should share common downstream genes. To assess this, we compared DEGs by overexpressing YAP or TAZ in myoblasts (Sun et al, 2017) with those of *Mpp7* cKO or *Amot* cKO cells, but found little overlap (Appendix Fig. S3E), presumably due to experimental differences. When next interrogated the promoters of DEGs from *Mpp7* or *Amot* cKOs cells, we found that most of them did harbor TEAD-binding sites (Fishilevich et al, 2017; Keenan et al, 2019), including that of *Carm1* (Appendix Fig. S3F). As several previous studies suggest some functional overlap

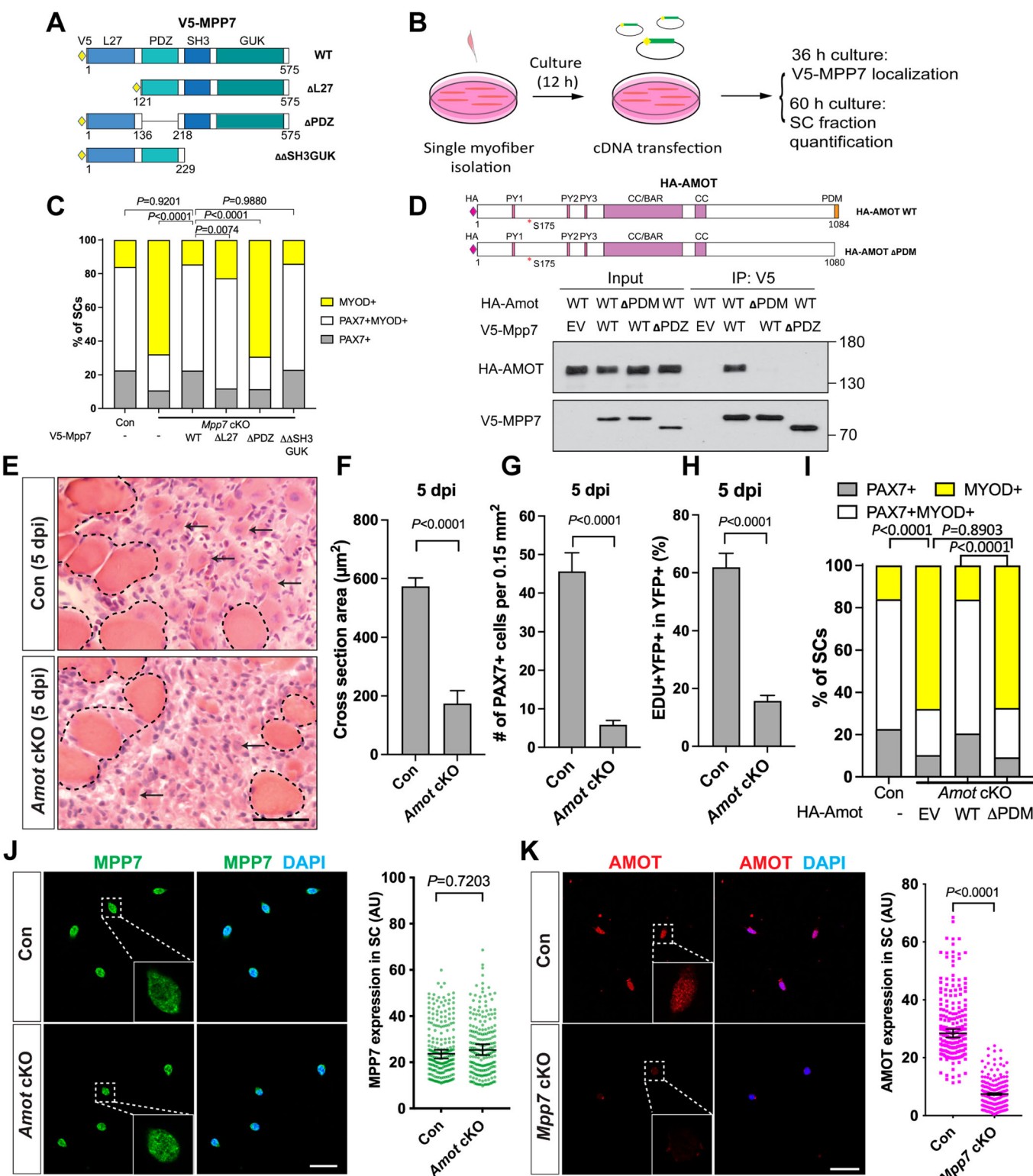

between YAP and TAZ, we assessed whether this might be the case in our studies. First, we confirmed that *Yap* cKO had compromised muscle regeneration (Sun et al, 2017), and found that *YapTaz* cKO had a severe defect with very few regenerated fibers (Fig. 4A). As *Taz* mutants had no regeneration defects (Sun et al, 2017), the

*YapTaz* cKO data indicated there is partial compensation for *Yap* function by *Taz*. Therefore, we next assessed DEGs in a *Yap* and *Taz* double cKO (*YapTaz* cKO, using *Taz^flox* and *Yap^flox* alleles (Reginensi et al, 2013)) and compared these to those found in the *Mpp7* and *Amot* cKO cells. The *YapTaz* cKO cells had 564 DEGs,

Figure 2. Interacting domains of MPP7 and AMOT are critical for SC renewal.

(A) Depiction of MPP7 domain architecture and V5-tagged wild type (WT) and domain deletion mutants (listed on the right) of Mpp7 expression constructs used in (B, C). (B) Flowchart to force-express various Mpp7 constructs (A) in Mpp7 cKO MuSCs. (C) Quantifications of cells fate fractions from experiments depicted in (B). IF of V5, PAX7 and MYOD was performed to determine the fate of transfected cells. Expression constructs used are in x-axis; (−), empty vector with IRES-GFP; keys to cell fate at the top. (D) Co-IP assays to determine interaction domains between MPP7 and AMOT in 293T cells. HA-tagged AMOT WT and AMOT △PDM, and V5-tagged MPP7 WT and MPP7 △PDZ were used in co-IP using an anti-V5 antibody, followed by Western blotting with anti-HA or anti-V5 antibodies. (E–H) Amot cKO regenerative defects at 5 dpi using the same experimental design as in Fig. 1B (Con, Pax7^CreERT2/+): Representative H&E staining images of TA muscles from Con (a different biological replicate compared to Fig. 1C, Con) and Amot cKO (E), quantifications of regenerated muscle fiber cross-sectional areas (F), PAX7+ MuSC densities (G), and percentages of EdU+ YFP-marked cells (H). (I) SM transfection assays (as depicted in B) with Amot WT or △PDM constructs; Con (−) from Fig. 2C; IF of PAX7, MYOD and HA was performed to determine transfected cells' fates; keys at the top. (J, K) IF of MPP7 in Con and Amot cKO MuSCs (J) and of AMOT in Con and Mpp7 cKO MuSCs (K), at 48 h after FACS isolation. Qualified fluorescent signals (arbitrary units, AU) are to the right. Data information: Scale bars = 25 μm (E, J, K). (C) ≥168 transfected cells were assessed per group; (F–H) N = 5 mice in each group; (I) ≥180 cells per group; (J, K) 200 cells per group. Bars represent means ± SD (F–H) or medians ± 95% CI (J, K). Student's t test (two-sided) were used in (F–H). Chi-square tests were performed in (C, I). Mann–Whitney test were performed in (J, K). Source data are available online for this figure.

relative to the control (Fig. 4B). As expected, cell growth and proliferation pathways were impacted (Appendix Fig. S4A). Thirty-three *Mpp7* cKO DEGs and 31 *Amot* cKO DEGs overlapped with *YapTaz* cKO DEGs, and 19 were common in all (Fig. 4B; Appendix Fig. S4B). TEAD binding sites are present in 18 out of 19 common DEG's promoters, including the *Carm1* promoter. A congruence of YY1 binding sites was also found in these promoters (Appendix Fig. S4C). Thus, MPP7 and AMOT likely act through YAP/TAZ (and YY1, see below) to regulate a selective set of target genes.

Neither *Yap* nor *Taz* was a DEG in *Mpp7* or *Amot* cKOs, but their protein levels were reduced in both cKOs cells; TAZ's level was the most affected in the *Mpp7* cKO (Fig. 4C,D). This suggests that reduced levels of YAP/TAZ in *Mpp7* and *Amot* cKOs are sufficient to maintain expression of most *Yap/Taz* downstream genes, and only a small number of *Yap/Taz* downstream genes is also dependent on *Mpp7* and *Amot* (i.e., common DEGs). Proteasome inhibitor MG132 restored the levels of TAZ, YAP, and AMOT in the *Mpp7* cKO near to those observed in the control cells. Curiously, CARM1 level was restored to < 50% of the control by MG132 (Appendix Fig. S4D). This suggests that in addition to normal levels of TAZ, YAP and AMOT, MPP7 is needed for a higher level of *Carm1* expression. Using TAZ as a representative (for TAZ and YAP), we found an enhanced interaction between MPP7 and TAZ by AMOT using co-IP assay (Fig. 4E; Appendix Fig. S4I). Furthermore, Mpp7 co-expression indeed increased Taz's transcriptional activity on a TEAD-reporter (Dupont et al, 2011) (Fig. 4F). Amot alone inhibited Taz, but Amot, Mpp7, and Taz altogether best activated the reporter. These results agree with AMOT's inhibitory role for TAZ(YAP), as well as with AMOT's positive role for TAZ(YAP) in a MPP7-dependent manner. Thus, MPP7 and AMOT can form a complex with TAZ and maximize TAZ's co-activator function. Along this line, we found that Carm1-reporter activation by Taz or Mpp7 is dependent on the TEAD binding site (Fig. 4G).

If the same mechanism applies to the MuSC, exogenous TAZ should not be able to activate high levels of *Carm1* nor rescue the renewal defect of the *Mpp7* cKO (i.e., without MPP7). We found that Taz and Taz S89A (a stabilized form of TAZ (Kanai et al, 2000)) could only increase CARM1 levels by ~2-fold (Fig. 4H; Appendix Fig. S4G) in *Mpp7* cKO cells (displaying ~5-fold reduced CARM1; Fig. 3G). Furthermore, TAZ S89A could only rescue the progenitor but not the renewal fraction of the *Mpp7* cKO (Fig. 4I). Yap and Yap S127A (stabilized form of YAP) also failed to rescue

the renewal fraction in this background (Appendix Fig. S4E). These data suggest that MPP7 (via AMOT) is needed for TAZ(YAP) to activate high levels of *Carm1* and possibly other common DEGs for renewal, whereas exogenous TAZ(YAP) can support the progenitor fate without MPP7.

We next ask whether Carm1 alone is sufficient to rescue *YapTaz* dKO renewal, as it did *Mpp7* cKO. Different from *Mpp7* cKO, *YapTaz* dKO SM culture contained mostly MyoD+Pax7- differentiating cells, few Pax7+MyoD+ progenitors, and no Pax7+-MyoD- cells (Appendix Fig. S4F). Carm1 partially rescued the progenitor and not the renewal fraction. Thus, other Yap/Taz downstream genes (at normal levels in *Mpp7* cKO) or Yap/Taz (present in *Mpp7* cKO) are needed together with Carm1 for renewal division (see "Discussion").

## The L27 domain of MPP7 enhances TAZ-mediated transcription and SC renewal

To decipher how MPP7 facilitates TAZ's activity via AMOT, we utilized mutant forms of MPP7 and TAZ. By co-IP, ΔPDZ MPP7 failed to interact with TAZ, ΔL27 MPP7 had a diminished interaction with TAZ, and ΔΔSH3GUK MPP7 had the same level of interaction with TAZ as WT MPP7 (Fig. 5A). ΔPBM AMOT interfered with MPP7-TAZ interaction (Appendix Fig. S5A) and TAZ with a mutated WW domain (TAZ WWm, unable to bind AMOT (Moleirinho et al, 2014)) failed to interact with MPP7 (Appendix Fig. S5B). As expected, TAZ WWm, not able to interact with AMOT and MPP7, showed lower transcriptional activity than TAZ in 293 T cells (Appendix Fig. S5C), and could partially rescue the progenitor but not the renewal fate in the *Mpp7* cKO (Fig. 4I). Thus, the interaction between MPP7 and TAZ is bridged by AMOT. It therefore followed that ΔPDZ Mpp7 (non-AMOT-binding) displayed little transcriptional activity on the TEAD-reporter (Fig. 5B). Unexpectedly, the ΔL27 Mpp7 also had little activity in the reporter assay, suggesting that the L27 domain of MPP7 (MPP7-L27) also contributes to enhancing TAZ activity when bridged by AMOT (diagram in Fig. 5C). Yet, ΔL27 Mpp7 could rescue the progenitor fraction in *Mpp7* cKO SM culture (Fig. 2C), suggesting a role in that context. We found that ΔL27 Mpp7 increased nuclear Yap/Taz (Appendix Fig. S4H). This helps to explain the rescue of progenitors by ΔL27 Mpp7, as Taz and Yap rescue progenitor but not renewal fractions in *Mpp7* cKO SM culture.

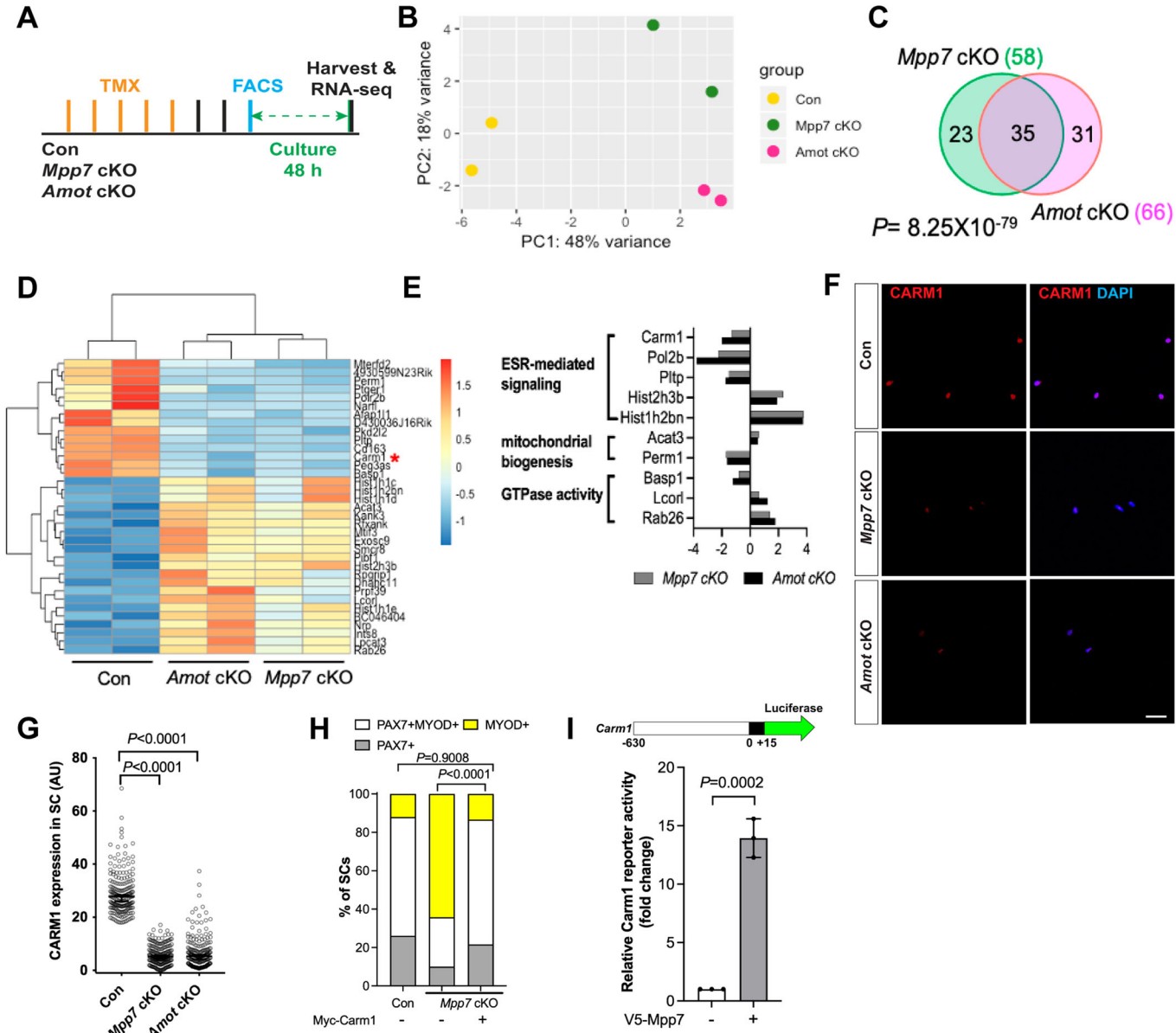

**Figure 3.  *Carm1* is commonly regulated by *Mpp7* and *Amot* in the MuSC.**

(A) Flowchart for bulk RNA-sequencing (RNA-seq). (B) PCA analysis of transcriptome data of Con (*Pax7^CreERT2/+*), *Mpp7* cKO, *Amot* cKO MuSCs. (C) Venn diagram summarizes (35) overlapping DEGs between the *Mpp7* cKO (58 DEGs) and the *Amot* cKO (66 DEGs). (D) Hierarchical clustering and heatmap of RNA-seq expression z-scores for the 35 overlapping DEGs; red asterisk, *Carm1*. (E) Expression changes (log2 FC, log2 fold change) of genes in GO-term enriched pathways (y-axis) are displayed for the *Mpp7* cKO (grey bars) and the *Amot* cKO (black bars). (F, G) Representative IF of CARM1 in Con (as in Fig. 1B), *Mpp7* cKO and *Amot* cKO MuSCs at 48 h in culture (F) and relative CARM1 fluorescent signals (in AU, G). (H) Expressing a Myc-tagged Carm1 rescued the *Mpp7* cKO in SM culture; Con (−) and *Mpp7* cKO (−) from Fig. 2C; IF of Myc, PAX7, and MYOD was performed to determine transfected cells' fates. Quantification of cell fate fractions is shown; keys at top. (I) V5-Mpp7 activated a Carm1-reporter (a luciferase reporter driven by a promoter region (−630 to +15) of *Carm1*, depicted at the top) in 293T cells; (−), empty expression construct. RNA-seq data deposit (NCBI) and analyses are in Methods. Data information: Scale bar = 25 μm in (F). (G) 200 MuSCs from 2 to 3 mice in each group; (H) ≥ 530 cells in each group; (I) *n* = 3 biological replicates. Bars represent medians ± 95% CI in (G) or means ± SD in (I). Hypergeometric test was used in (C). Kruskal–Wallis test followed by Dunn's multiple comparisons test was used in (G), Chi-square test in (H), and Student's *t* test (two-sided) in (I). Source data are available online for this figure.

If the suggested role for MPP7-L27 above is correct, a fusion of MPP7-L27 to TAZ (L27-TAZ; Fig. 5D) should have a higher transcriptional activity than TAZ. Indeed, L27-Taz activated the TEAD-reporter to the same level as Taz and Mpp7 together (Fig. 5E). We dissected MPP7-L27, which contains L27N and L27C repeats. L27N-Taz and L27C-Taz had slightly reduced transcriptional activities

than L27-Taz (Appendix Fig. S5D). Lysine 38 (L38) in L27N and L95 in L27C mediate interactions with DLG and LIN7, respectively (Bohl et al, 2007). Mutating either lysine did not compromise the activity of L27-Taz, revealing that DLG and LIN7 were not involved. Importantly, L27-Taz was more efficient than Taz in increasing CARM1 levels (Appendix Fig. S5E) and sufficient to rescue both progenitor and

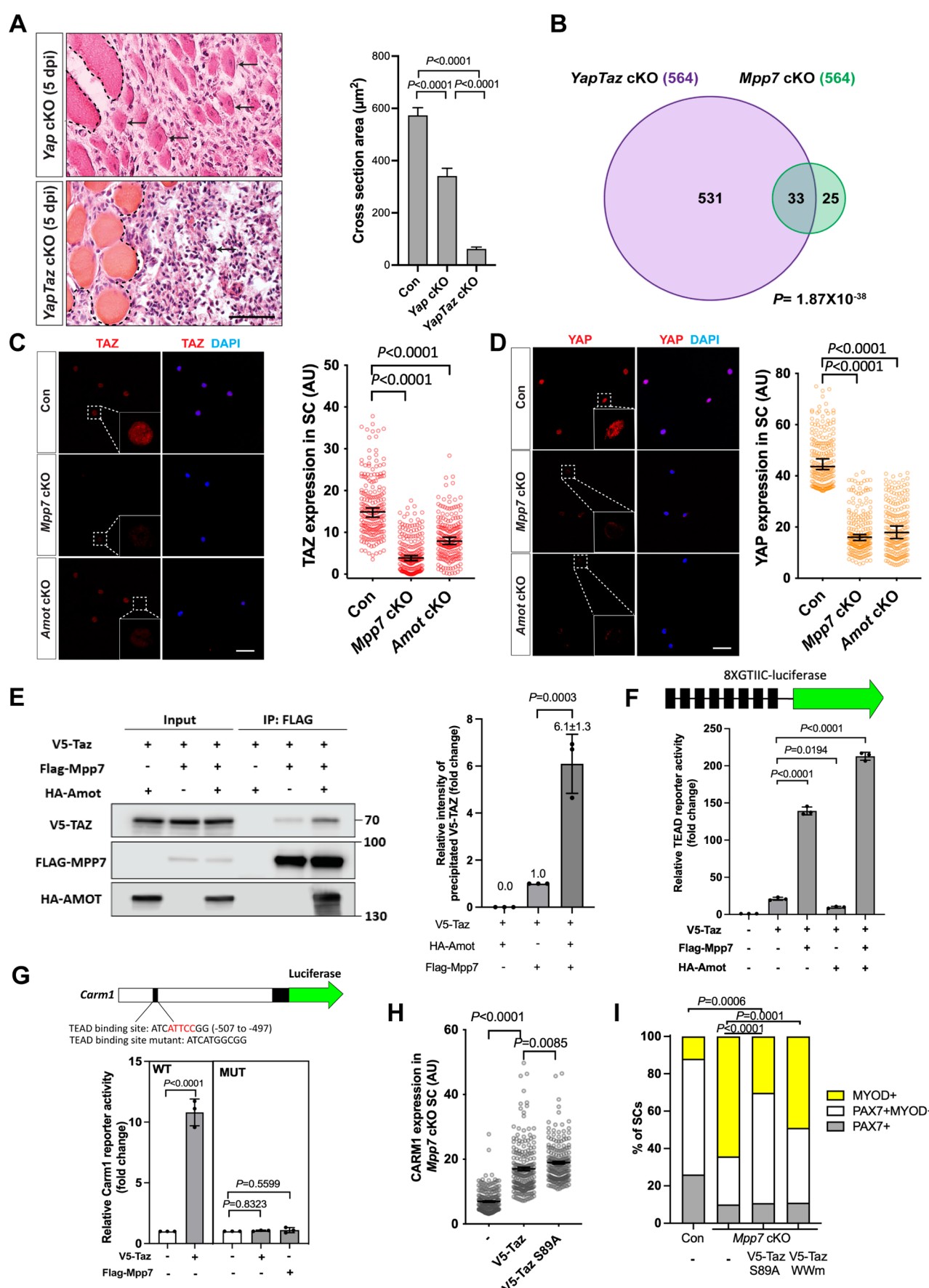

Figure 4. *Mpp7/Amot* regulatory network intersects with that of *Yap/Taz.*

(A) H&E histology of *Yap* cKO and *YapTaz* cKO muscles at 5 dpi (Con (*Pax7^CreERT2/+*) histology not included); quantifications of regenerated myofiber cross-sectional area to the right. (B) Venn diagram shows overlapping DEGs between the *YapTaz* cKO and the *Mpp7* cKO. (C, D) IF images of TAZ (C) and YAP (D) in FACS-isolated and cultured Con (*Pax7^CreERT2/+*), *Mpp7* cKO, and *Amot* cKO MuSCs at 48 h; relative fluorescent signals (in AU) to the right. (E) Co-IP of V5-TAZ and HA-AMOT by FLAG-MPP7 expressed in 293 T cells; tagged epitopes used for IP and Western blotting are indicated; (−), empty expression construct. Quantification of relative levels of co-IPed V5-TAZ is to the right. (F) Relative TEAD-reporter (8XGTIIC-luciferase, depicted at top) activities when co-transfected with V5-Taz, Flag-Mpp7, and/or Ha-Amot constructs in 293T cells; (−), empty expression construct. (G) Relative activities of WT and TEAD-binding site mutated (MUT) Carm1-reporters co-transfected with V5-Taz or Flag-Mpp7 constructs; (−), empty expression construct. (H) Relative IF signals (AU) of CARM1 in *Mpp7* cKO MuSCs transfected with empty vector (−), V5-Taz WT and V5-Taz S89A constructs (all with IRES-mGFP) (I) Relative cell fate fractions among Con (*Pax7^CreERT2/+*), *Mpp7* cKO, and *Mpp7* cKO MuSCs transfected with V5-Taz S89A and V5-Taz WWm construct in single myofiber culture; Con (−) and *Mpp7* cKO (−) from Fig. 2C; keys at the top. Data information: Scale bar = 50 μm in (A) and 25 μm in (C, D). (A) $N = 5$ mice in each group; (C, D, H) 200 cells from 2 to 3 mice in each group; (E–G) $n = 3$ biological experiments; (I) ≥150 cells in each group. Bars represent medians ± 95% CI (C, D, H) or means ± SD (A, E, F, G). Hypergeometric test was used in (B), one-way ANOVA with Tukey's post hoc test performed in (A, E, F, G), Kruskal–Wallis test followed by Dunn's multiple comparisons test in (C, D, H), and Chi-square test used in (I). Source data are available online for this figure.

renewal fates of the *Mpp7* cKO (Fig. 5F). Having established the role of MPP7-L27, we asked two additional questions: (1) Is AMOT's role solely to bridge TAZ (YAP) with MPP7? (2) How does MPP7-L27 enhance TAZ function?

## AMOT acts as a F-actin-regulated shuttling factor in MuSCs

Cellular distribution of AMOT, MPP7, and YAP in MuSCs is dynamic, from the apical surface to the nucleus in SM culture (Li and Fan, 2017). We confirmed those observations by co-IF with M-cadherin or PAX7 antibodies in 3D reconstructed images (Appendix Fig. S6A–F). As MuSC activation is accompanied by actin rearrangement (Kann et al, 2022) and AMOT can bind F-actin (Ernkvist et al, 2006), we asked whether nuclear AMOT is regulated by actin polymerization states. For this, we subjected isolated MuSCs to pharmacological manipulations of F-actin (Fig. 6A). Immediately after plating, MuSCs treated with the F-actin stabilizers Jasplakinolide (Jasp) and Narciclasine (Nar) showed increased nuclear localization of AMOT and MPP7, compared to mock-treated cells (Fig. 6B,C). Conversely, MPP7 and AMOT were mostly nuclear at 48 h of culture but shuttled out to the cytoplasm when treated with Blebbistatin (Bleb), Cytochalasin B (Cyto B), or Y-27632 to weaken/disrupt F-actin (Fig. 6D,E).

MPP7 and AMOT did not show strict co-localization, but their compartmental distributions were similarly regulated by F-actin states. Given that AMOT stability depends on MPP7, their transient/dynamic interaction may be sufficient for coordinated localization in mass. Alternatively, MPP7 and AMOT have other partners (e.g., other AMOT or MPP family members) and those complexes are similarly regulated by F-actin states. Either way, their coordinated cellular distribution likely lies in AMOT-F-actin interaction (Ernkvist et al, 2006), which is modulated by phosphorylation of serine 175 (S175): S175A AMOT can bind actin and phospho-mimetic S175E AMOT cannot (Chan et al, 2013; Dai et al, 2013). We found that higher percentages of WT and S175A AMOT were present in the nucleus compared to S175E AMOT when expressed in the *Amot* cKO cells (Fig. 6F). Unlike WT Amot, S175A Amot could only rescue the progenitor fate and S175E Amot had no rescue activity (Fig. 6F). S175A Amot has been shown to elevate nuclear YAP (Moleirinho et al, 2017; Yi et al, 2013), which helps explain the rescue of progenitor fate. As AMOT-PDM (MPP7-binding) is critical for renewal, we examined whether AMOT's binding to TAZ/YAP is also required. AMOT

with all 3 TAZ/YAP binding motifs mutated (denoted as 3PY) was largely cytoplasmic and did not rescue *Amot* cKO renewal (Fig. 6G). Altogether, the dynamic association of AMOT with F-actin instead of a strictly phosphorylated or non-phosphorylated form of AMOT per se, is critical for MuSC renewal, and its binding to both MPP7 and YAP/TAZ is indispensable.

## YY1 adds another dimension to MPP7-L27 and TAZ for transcriptional activity

As mentioned above, YY1 binding sites are prevalent in the promoters of commonly shared DEGs from the *Mpp7*, *Amot*, and *YapTaz* cKOs cells. *Yy1* cKO mice had severe muscle regeneration defects and DEGs in the categories of mitochondrial and glycolysis genes (Chen et al, 2019). We found a small but significant overlap between our cKOs' and *Yy1* cKO's DEGs (Appendix Fig. S7A); four of these overlapping DEG promoters are shown in Fig. 7A.

As for function, we showed that either Yy1 or Taz expression could activate the Carm1-reporter. L27-Taz had a higher activity than Taz with or without Yy1, but no synergy was observed (Fig. 7B). Mutating either TEAD or YY1 binding site abolished reporter activation by any combination of Yy1, Taz, and L27-Taz (Fig. 7C,D); WT Mpp7 also could not activate mutated reporters (Appendix Fig. S7B). A likely explanation is that exogenous YY1 or TAZ activate the promoter by cooperating with endogenous TAZ(YAP) and YY1, respectively. That is, their ubiquitous expression masks their cooperativity, and co-existing binding sites endow their co-occupancy for activation.

As far as promoter co-occupancy, it is possible that they interact with each other. Indeed, TAZ and YY1 could interact with each other by co-IP, and L27-TAZ performed better than TAZ (Fig. 7E). YY1 likely interacted with MPP7 indirectly via endogenous AMOT, as their interaction required the PDZ domain of MPP7 (Fig. 7F). MPP7's interaction with YY1 (through endogenous AMOT) was diminished when the L27 domain was deleted, similar to its interaction with TAZ. To exclude the contribution of AMOT to Carm1-reporter activation by L27-TAZ - as TAZ can recruit AMOT, we showed that the L27-Taz WWm (non-AMOT binding) activated the reporter equally well as L27-Taz, with or without YY1 (Appendix Fig. S7C). Lastly, when Mpp7-L27 was fused to the Gal4-DNA-binding domain (Tang et al, 2013), it activated a Gal4-UAS-reporter (Potter et al, 2010) (Appendix Fig. S7D), revealing an intrinsic activator function. Thus, MPP7, AMOT, TAZ(YAP), and YY1 form a strong transcription activator complex (Fig. 7G)

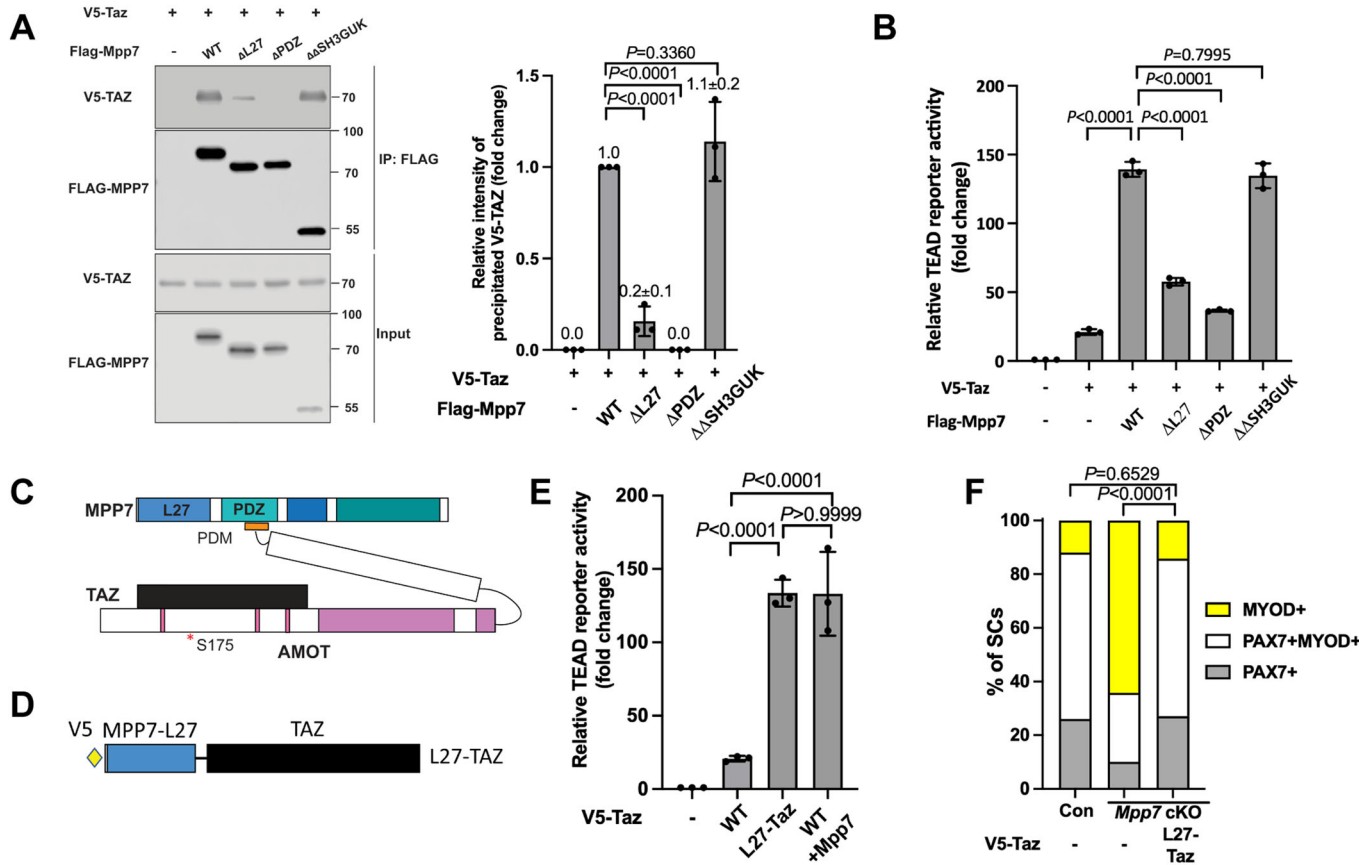

**Figure 5. AMOT links TAZ to the MPP7-L27 domain to enhance transcription.**

(A) MPP7-L27 contributes to MPP7-TAZ interaction by co-IP assay in 293T cells. Expression constructs and tagged epitopes for detection are indicated; (−), empty vector; quantification to the right. (B) Relative TEAD-reporter activities when co-transfected with various Mpp7 constructs (x-axis); (−), empty vector. n = 3 individual experiments. (C) Model summarizes the co-IP results in (A, Appendix Fig. S5A,B). (D) Diagram for the fusion construct of V5-tagged Mpp7-L27 and Taz (i.e., L27-Taz) used in (E, F). (G) Relative TEAD-reporter activities when co-transfected with constructs indicated in the x-axis; (−), empty expression construct. (H) Quantification of cell fates of Con and Mpp7 cKO MuSCs in SM assay; expression constructs in the x-axis; Con (−) and Mpp7 cKO (−) from Fig. 2C. Data information: (A, B, E) n = 3 biological replicates; (F) ≥ 150 cells in each group. Error bars represent means ± SD in (A, B, E). One-way ANOVA with Tukey's post hoc test was performed in (A, B, E), and Chi-Squire test in (F). Source data are available online for this figure.

combining multiple activation domains (of TAZ, YY1 and MPP7) to activate *Carm1*-reporter expression at a high level. Although AMOT does not appear to contribute to transcriptional activation per se, it holds the complex together.

## MPP7, TAZ, and YY1 converge onto the Carm1 promoter to support MuSC renewal

The above results predict co-occupancy on common DEGs' promoters by MPP7, TAZ(YAP) and YY1. To assess this, we performed chromatin immunoprecipitation–quantitative PCR (ChIP-qPCR) using primary myoblasts. MPP7, TAZ and YY1 were bound to the proximal promoter region with TEAD and YY1 bindings sites in the *Carm1* promoter, but not at a distal region where TEAD or YY1 binding sites are not present (Fig. 8A). We obtained similar results at additional common DEG promoters (Fig. 8B). In contrast, promoters of *Yy1* cKO-only DEGs with YY1 binding sites were occupied by YY1 but not TAZ, and promoters of *YapTaz1* cKO-only DEGs with TEAD binding sites were occupied by TAZ but not YY1 (Appendix Fig. S8A,B). MPP7 occupancy was

not detected at non-common DEG promoters, consistent with MPP7's selectivity in regulating common DEGs through co-existing TEAD and YY1 binding sites.

We next employed PLA to demonstrate interactions between endogenous proteins of interest. PLA signals were observed between pairs of endogenous MPP7-TAZ/YAP, MPP7-YY1, and TAZ/YAP-YY1 in MuSCs at 48 h in SM cultures (Fig. 8C–E; control and AMOT-MPP7 PLA images in Appendix Fig. S8C,D). Noteworthy, PLA signals for YY1-TAZ/YAP were drastically reduced in the *Mpp7* cKO, compared to those in the control (Fig. 8E), consistent with a combination of MPP7 loss and TAZ reduction. PLA signals of CARM1 and PAX7 (as an indicator for PAX7 methylation and renewal (Kawabe et al, 2012)) were diminished in the *Mpp7* cKO cells, in line with the defect in renewal (Fig. 8F). Importantly, L27-Taz expression, which rescued the renewal defects of *Mpp7* cKO MuSCs, was able to increase PLA signals of CARM1 and PAX7 in the *Mpp7* cKO (Fig. 8G). Together with the ChIP-qPCR data, these results further strengthen our model for converged actions of MPP7, AMOT, TAZ(YAP), and YY1 to regulate a select set of genes to drive MuSC renewal.

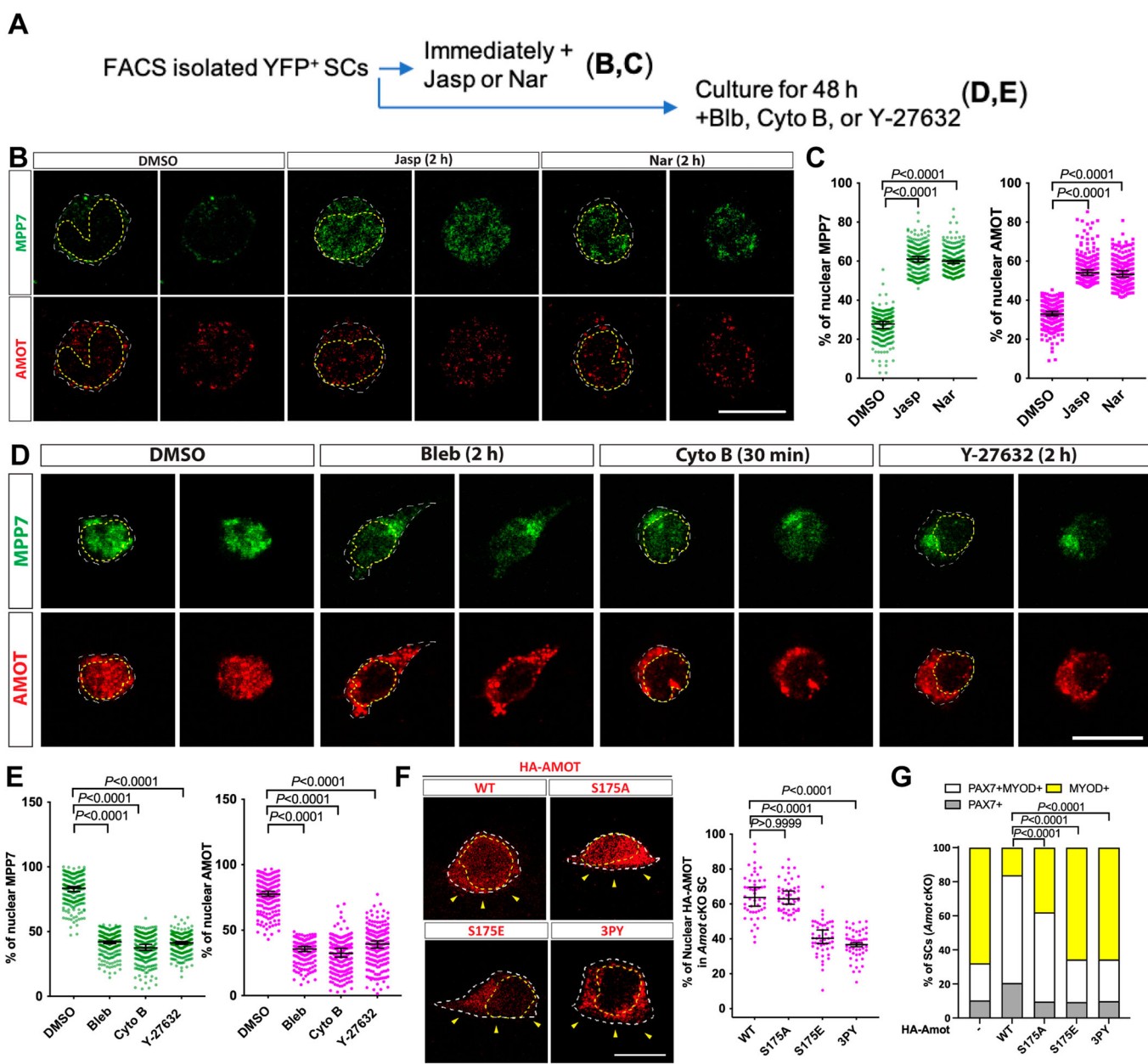

**Figure 6. F-actin states impact on AMOT localization and function through TAZ/YAP.**

(A) Experimental flowchart to investigate the impact of F-actin on the localization of MPP7 and AMOT in FACS-isolated MuSCs. (B) Representative IF images of MPP7 and AMOT of freshly isolated MuSCs treated with DMSO (control), 100 nM Jasplankinolide (Jasp), or 100 nM Narciclasine (Nar) for 2 h. (C) Percentages of MPP7 or AMOT IF signals in the nucleus (versus total signals) in each SC were quantified. (D) Representative IF images of MPP7 and AMOT in MuSCs cultured for 48 h and then treated with DMSO (control), 10 μM Blebbinstatin (Bleb), 10 μM Cytochalasin B (Cyto B), or 10 μM Y-27632 for indicated time prior to assay; (E) quantification same as in (C). (F) Localization of HA-tagged AMOT WT, AMOT S175A (S175A), AMOT S175E (S175E), and AMOT 3PY (3PY) expressed (via transfection) in *Amot* cKO MuSCs on SM by IF for HA; yellow arrowheads, apical side. Percentages of nuclear signals (of total signal) of each variant were quantified. (F) HA-AMOT variants in (G) were transfected into *Amot* cKO MuSCs on SM; *Amot* cKO (-/WT) from Fig. 2I. Cell fates of transfected cells were determined by IF for HA, PAX7 and MYOD and quantified. Data information: Scale bars = 10 μm in (B, D). (C, E) 200 MuSCs in each group; (F) 50 MuSCs in each group; (G) ≥ 145 cells in each group. Bars represent medians ± 95% CI in (C, E, F) and means ± SD in (F). Kruskal–Wallis test followed by Dunn's multiple comparisons test in (C, E, F) and Chi-square test in (G). Source data are available online for this figure.

## Discussion

Using complementary approaches, we show here that MPP7, AMOT, TAZ(YAP), and YY1 form a complex that converges to regulate common DEGs, e.g., *Carm1*, to drive MuSC renewal. Their convergence relies on co-existing TEAD and YY1 binding sites and the L27-domain of MPP7 leading to elevated transcriptional output. The limited number of common DEGs might suggest that a regulator pathway involving *Amot*, *Mpp7*, *Taz/Yap*, and *Yy1* play a secondary role. Alternatively, given their diverse roles in the

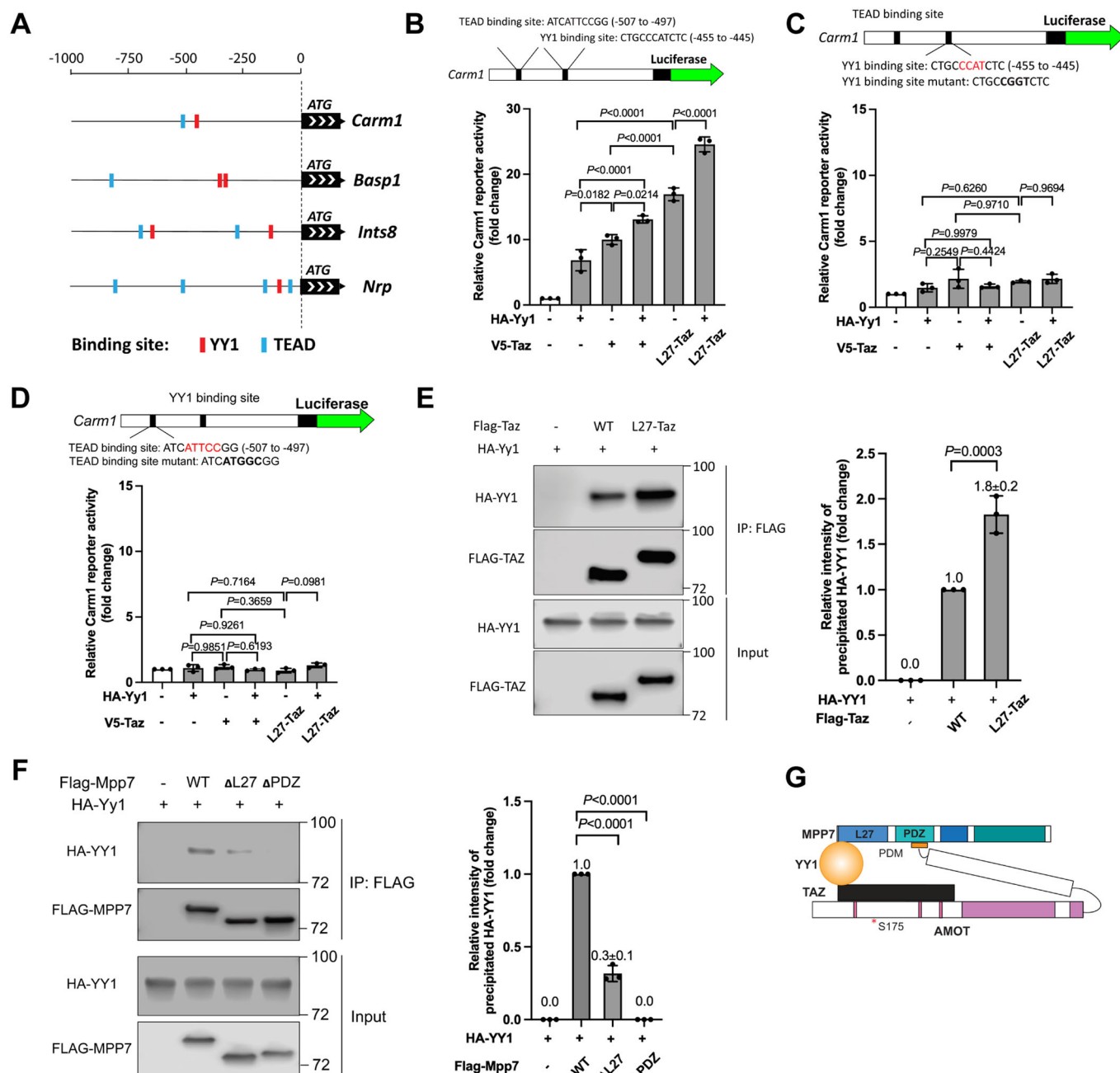

**Figure 7. The L27 domain of MPP7 cooperates with TAZ and YY1 to enhance transcription.**

(A) Promoters of 4 common DEGs among *Mpp7, Amot, YapTaz*, and *Yy1* cKO (Chen et al, 2019) data sets. Putative YY1 binding sites are red blocks and TEAD binding sites, blue blocks. (B) Location and sequence of the TEAD and YY1 binding sites are indicated in the *Carm1* promoter. Carm1-reporter was co-transfected with constructs in x-axis to assess transcriptional activity; (−), empty expression construct. (C) Same as in (B), except that the YY1 binding site is mutated (top). (D) Same as in (B), except that the TEAD binding site was mutated (top). (E) Co-IP between TAZ and YY1 is enhanced by fusing MPP7's L27 domain to TAZ in 293T cells. Constructs and tagged epitopes for detection are indicated; quantification of co-IPed HA-YY1 to the right. (F) Co-IP assay to determine the relative contributions of L27 and PDZ domains of MPP7 to its interaction with YY1 in 293T cells. Constructs and tagged epitopes for detection are indicated; quantification of co-IPed HA-YY1 to the right. (G) A model for TAZ-YY1 cooperation mediated by AMOT and MPP7. S175 of AMOT is subjected to phosphorylation and regulation by actin dynamics. Data information: (B–F) $n = 3$ biological replicates. Error bars represent means ± SD. One-way ANOVA with Tukey's post hoc test was performed in (B–F). Source data are available online for this figure.

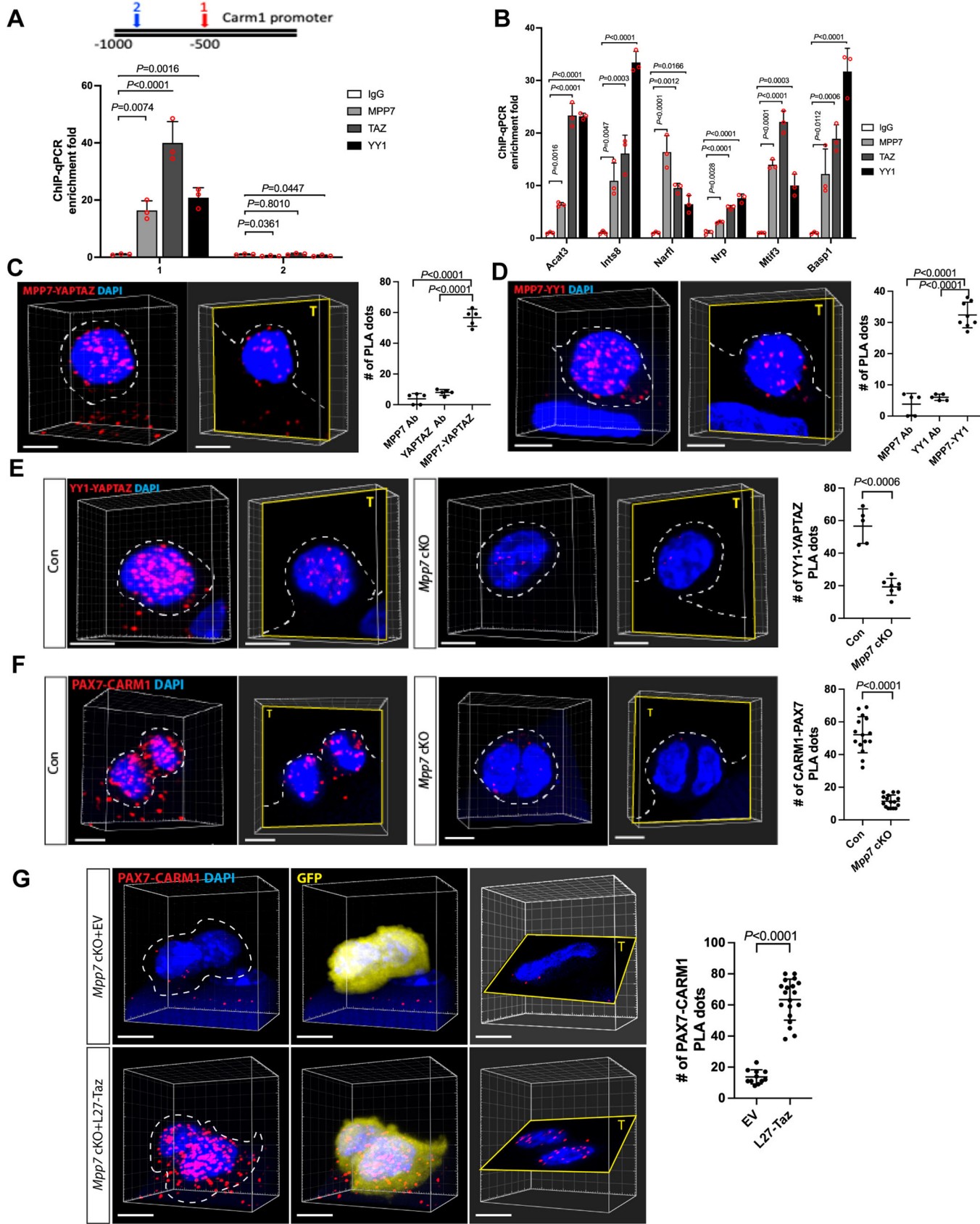

**Figure 8. Chromatin binding and protein association support convergence of MPP7, YY1, and TAZ/YAP.**

(A) ChIP-qPCR shows MPP7, YY1, and TAZ binding to *Carm1* promoter (diagramed a top) at site 1 (red) harboring both TEAD and YY1 binding sites, but not at a distal site 2 (blue) without either binding site; IgG, negative control. Bar graph shows fold-enrichment (by Bio-Rad CFX Maestro); keys to the left. (B) Same as in (A) with 6 more common DEG promoters (x-axis) with YY1 and TEAD binding sites; keys and quantification as in (A). (C) Representative PLA images for MPP7 and YAP/TAZ in 3D reconstruction and single plane, countered stained with DAPI; quantification to the right. (D) Same as in (C) for MPP7 and YY1. (E) Same as in (C) for YY1 and YAP/TAZ in Con (*Pax7*$^{CreERT2/+}$) and *Mpp7* cKO. (F) Same as in (E) for CARM1-PAX7; (G) Restoring CARM1-PAX7 PLA signal by L27-TAZ but not by empty vector (EV, with IRES-mGFP, same as (−) in previous figures). Each data point in (C–G) represents PLA dots in a cell quantified by 3D imaging. Single Ab controls are shown in Appendix Fig. S8C. Data information: Scale bars = 5 μm (C–G). (A, B) n = 3 biological replicates. (C) 5 MuSCs for each group; (D) 5 MuSCs for MPP7 or YY1 and 8 MuSCs for MPP7-YY1; (E) 5 MuSCs for Con (*Pax7*$^{CreERT2/+}$) and 7 MuSCs for *Mpp7* cKO; (F) 15 MuSCs for Con and 16 MuSCs for *Mpp7* cKO; (G) 11 MuSCs for EV and 19 MuSCs for L27-Taz. Error bars represent means ± SD. One-way ANOVA with Tukey's post hoc test in (A–D). Student's *t* test (two-sided) in (E–G). Source data are available online for this figure.

literature, the common DEGs targeted by these effectors represent a precise and tightly regulated function that requires several inputs to activate. With this in mind, we designed experiments to assess their convergent functions. We chose *Carm1* as a model DEG to build our case due to its documented role in MuSC renewal cell division, and extended the ChIP data to other DEG promoters. Whilst we show that MuSC renewal and CARM1-PAX7 interaction are restored by L27-Taz (in the *Mpp7* cKO), CARM1 may potentially methylate other proteins that participate in renewal. Carm1 alone is sufficient to rescue renewal fraction in *Mpp7* cKO but not *YapTaz* dKO, implying that certain *Yap/Taz* downstream gene(s) not affected in *Mpp7* cKO is needed for Carm1-mediated renewal division. We suggest that such co-required Yap/Taz downstream genes are involved in cell growth/cell cycle as a pre-requisite for Carm1-mediated renewal division. How Carm1 helps to expand the progenitor fraction in *Mpp7* cKO and *YapTaz* dKO is unknown but likely involves methylation target(s) other than Pax7. Lastly, not all Carm1 transfected cells take on the renewal fate, clearly showing additional layers of regulation to direct the relative ratios of MuSC-derived fates. Future studies are needed to answer these intriguing open questions. Worth noting is that CARM1 has also been documented to potentiate myogenic differentiation via distinct interacting proteins (Chen et al, 2002).

In addition to Carm1, other common DEGs may also contribute to renewal or other functional aspects of MuSCs. Conversely, partial-overlapping DEGs from various cross-comparisons of *Mpp7*, *Amot*, *Yap/Taz* and/or *Yy1* cKOs hint at intriguing combinatorial codes for different functional outputs. Combinatorial and convergent regulatory networks are likely the inherent nature of a critical biological process such as stem cell renewal which require several layers of regulation. Our synthesis of a hand full of players is but one step towards the path of understanding MuSC renewal. Future studies to incorporate G-actin sensor MRTF (Kann et al, 2022), other regulators of YAP/TAZ (Rausch and Hansen, 2020), and other known players in MuSC renewal (Relaix et al, 2021; Sousa-Victor et al, 2022) should help provide a comprehensive view.

We propose that the logic of a tiered control of *Carm1* expression levels, in addition to its post-transcriptional regulation (Chang et al, 2018), lies in a convergent checkpoint of YY1's and YAP/TAZ's transcriptional programs. YY1 largely controls mitochondrial and glycolytic genes (Chen et al, 2019), whereas YAP/TAZ largely controls cell growth and proliferation genes (Sun et al, 2017). These cellular functions must be coordinated for robust stem cell activation and progenitor expansion. Once these cellular conditions are matched, high-level activation of renewal genes, e.g. *Carm1*, is set in motion by convergence. Inclusion of AMOT and MPP7 not only adds to transcriptional enhancement, but also provides a layer of mechano-checkpoints for MuSCs to sense their local physical environment and to adjust the renewal rate accordingly. Lastly, both YY1

and YAP/TAZ pathways have also been extensively studied in cancer cells. Whether *Mpp7* and *Amot* also contribute to cancers originating from MuSCs, e.g., rhabdomyosarcoma, deserves future attention.

## Methods

### Reagents and tools table

| Reagent/resource | Reference or source | Identifier or catalog number |
|---|---|---|
| **Experimental models** | | |
| Pax7$^{CreERT2}$ (*M. musulus*) | Jackson Lab (Lepper et al, 2009) | Stock #: 012476 |
| Rosa26$^{YFP}$ (*M. musulus*) | Jackson Lab (Srinivas et al, 2001) | Stock #: 006148 |
| Mpp7$^{flox}$ (*M. musulus*) | This paper | Available upon request |
| Amot$^{flox}$ (*M. musulus*) | from Dr. Joseph L Kissil (Shimono and Behringer, 2003) | |
| Yap$^{flox}$Taz$^{flox}$ (*M. musulus*) | Jackson Lab (Reginensi et al, 2013) | Strain #: 030532 |
| **Recombinant DNA** | | |
| pCDNA3-V5-Mpp7 (human cDNA) | This paper | Available upon request |
| pCDNA3-V5-Mpp7 △L27 | Modified from pCDNA3-V5-Mpp7; This paper | Available upon request |
| pCDNA3-V5-Mpp7 △PDZ | Modified from pCDNA3-V5-Mpp7; This paper | Available upon request |
| pCDNA3-V5-Mpp7 △△GukSH3 | Modified from pCDNA3-V5-Mpp7; This paper | Available upon request |
| pCDNA3-Flag-Mpp7 | Modified from pCDNA3-V5-Mpp7; This paper | Available upon request |
| pCDNA3-Flag-Mpp7 △L27 | Modified from pCDNA3-Flag-Mpp7; This paper | Available upon request |
| pCDNA3-Flag-Mpp7 △PDZ | Modified from pCDNA3-Flag-Mpp7; This paper | Available upon request |
| pCDNA3-Flag-Mpp7 △△SH3GUK | Modified from pCDNA3-Flag-Mpp7; This paper | Available upon request |
| HA-Amot p130 (human cDNA) | Addgene (Zhao et al, 2011) | Catalog #32821 |
| HA-Amot p130 △PDM | From Dr. Joseph L Kissil (Moleirinho S et al, 2017) | |
| HA-Amot p130 S175A | From Dr. Joseph L Kissil (Moleirinho S et al, 2017) | |

| Reagent/resource | Reference or source | Identifier or catalog number |
|---|---|---|
| HA-Amot p130 S175E | From Dr. Joseph L Kissil (Moleirinho S et al, 2017) | |
| HA-Amot p130 3PY | From Dr. Joseph L Kissil (Moleirinho S et al, 2017) | |
| Flag-Taz (human cDNA) | From Dr W Hong (Chan et al, 2011) | |
| Flag-Taz wwm (W152A, P155A) | From Dr W Hong (Chan et al, 2011) | |
| Myc-Carm1 (human cDNA) | Origene | Catalog #RC217483 |
| HA-YY1 (human cDNA) | Addgene (Weintraub et al, 2017) | Catalog #104395 |
| Flag-L27-Taz | Modified from pCDNA3-Flag-Mpp7 and Flag-Taz; This paper | Available upon request |
| pGL4-Carm1 | This paper | Available upon request |
| 8xGTIIC-luciferase | Addgene (Dupont et al, 2011) | Catalog #34615 |
| **Antibodies** | | |
| Rabbit anti-MPP7 | Proteintech | Catalog #12983-1-AP |
| Mouse anti-AMOT IgG2b | Santa Cruz | Catalog #sc-166924 |
| Mouse anti-PAX7 IgG1 | Developmental Studies Hybridoma Bank | Catalog #PAX7, Registration ID: AB_528428 |
| Rabbit anti-MYOD | Santa Cruz | Catalog #sc-304 |
| Chicken anti-LAMININ | Antibodiesonline.com | Catalog #ABIN573807 |
| Mouse anti-N-Cadherin IgG1 | Santa Cruz | Catalog #sc-393933 |
| Mouse anti-M-Cadherin IgG1 | Santa Cruz | Catalog #81471 |
| Mouse anti-beta-catenin | Santa Cruz | Catalog #sc-7963 |
| Rabbit anti-PAR3 | Millipore | Catalog #07-330 |
| Rabbit anti-CARM1 | Bethyl | Catalog #IHC-00045 |
| Rabbit anti-YAP | Cell Signaling | Catalog #14074 |
| Rabbit anti-TAZ | Cell Signaling | Catalog #83669 |
| Mouse anti-YY1 | Santa Cruz | Catalog #sc-7341 |
| Rabbit anti-FLAG | Cell Signaling | Catalog #14793 |
| Rabbit anti-HA | Cell Signaling | Catalog #3724 |
| Mouse anti-HA | Cell Signaling | Catalog #2367 |
| Chicken anti-GFP | Aves Lab | Catalog #GFP-1020 |
| Chicken anti-c-MYC Tag | Bethyl | Catalog #A190-103A |
| Rabbit anti-V5 | Cell Signaling | Catalog #13202 |
| Goat anti-mouse IgG1 cross-adsorbed secondary antibody, Alexa 568 | Thermo Fisher Scientific | Catalog #A-21123 |
| Goat anti-rabbit IgG (H + L) cross-adsorbed secondary antibody, Alexa 568 | Thermo Fisher Scientific | Catalog #A-11011 |
| Goat anti-mouse IgG2b cross-adsorbed secondary antibody, Alexa 568 | Thermo Fisher Scientific | Catalog #A-21144 |
| Goat anti-chicken IgY Secondary Antibody, FITC | Aves Lab | Catalog #F-1005 |
| Anti-V5 Agarose | Millipore Sigma | Catalog #A7345 |
| Anti-FLAG M2 magnetic beads | Millipore Sigma | Catalog #M8823 |
| **Oligonucleotides and sequence-based reagents** | | |
| Carm1-1-ChIP-qPCR | Forward: cattccggggggcgtgc Reverse: aggcgctttgtgccacc | |
| Carm1-2-ChIP-qPCR | Forward: ccgtcccttgacaaaaa gatgc Reverse: cccaggagggacggttacta | |
| Acat3-ChIP-qPCR | Forward: gtcccggctgaatcatcaga Reverse: tcccttttctgtctgttttttgtgt | |
| Ints8-ChIP-qPCR | Forward: cgaagacatcgaactcgctt Reverse: tagattctggcggggctct | |
| Narfl-ChIP-qPCR | Forward: agggaaactgggaaaggg gat Reverse: ttgccaggaggattcttgtttt | |
| Nrp-ChIP-qPCR | Forward: cagtgcgcttagcccccttta Reverse: cacgactccagggtttcgat | |
| Mtif3-ChIP-qPCR | Forward: tggataccatgtgggtgctg Reverse: tggcccagagggttaagagtc | |
| Basp1-ChIP-qPCR | Forward: agttctaaaatggctgtcc ctg Reverse: atccaggaggcttgaacacc | |
| Ankrd1-ChIP-qPCR | Forward: aaaaagggcagtgatgtg gtg Reverse:gaagagggagggggaggacaa | Zanconato et al, 2015 |
| Amotl2-ChIP-qPCR | Forward: tgccaggaatgtgagagtttc Reverse: aggaggggagcgggagaag | Zanconato et al, 2015 |
| Mrpl11-ChIP-qPCR | Forward: ttaccctagccgaacacgag Reverse: cttagctcgcctcggagaag | Chen et al, 2019 |
| Uqcrh-ChIP-qPCR | Forward: ctgctcctctgtttgacgat Reverse: agaggtcagcttttaggaccg | Chen et al, 2019 |
| **Chemicals, enzymes and other reagents** | | |
| Cardiotoxin | Millipore Sigma | Catalog #11061-96-4 |
| Tamoxifen | Millipore Sigma | Catalog #10540-29-1 |
| EdU(5-ethynyl-2′-deoxyuridine) | Millipore Sigma | Catalog #61135-33-9 |
| 4-hydroxytamoxifen (4-OH-TMX) | Tocris Bioscience | Catalog #3412 |
| Collagenase, Type 2 | Worthington Biochemical | Catalog #LS004176 |

| Reagent/resource | Reference or source | Identifier or catalog number |
|---|---|---|
| Dispase II | Thermo Fisher Scientific | Catalog #17105041 |
| Matrigel | Corning | Catalog #354234 |
| GlutaMax | Thermo Fisher Scientific | Catalog #A1286001 |
| Chicken embryo extract | MP Biomedicals | Catalog #MP92850145 |
| FGF2 | R&D Systems | Catalog #3718-FB-010 |
| TransfeX Reagent | ATCC | Catalog #ACS-4005 |
| Lipofectamine 3000 | Thermo Fisher Scientific | Catalog #L3000008 |
| Y-27632 | Tocris Bioscience | Catalog #1254 |
| Blebbinstatin | Tocris Bioscience | Catalog #1852 |
| Cytochalasin B | Tocris Bioscience | Catalog #5474 |
| Narciclasine | Tocris Bioscience | Catalog #3715 |
| Jasplakinolide | Tocris Bioscience | Catalog #2792 |
| TRIzol LS Reagent | Thermo Fisher Scientific | Catalog #10296010 |
| Mouse on Mouse (M.O.M) blocking reagent | Vector Laboratories | Catalog #BMK-2202 |
| Carbo-free blocking solution | Vector Laboratories | Catalog #SP-5040-125 |
| BD insulin syringe | BD | Catalog #324792 |
| PFA | Electron Microscopy Sciences | Catalog #15710-SP |
| FSC 22 frozen section media | Leica | Catalog #3801480 |
| Isopentane | VWR | Catalog #MK196759 |
| Normal goat serum | Gibco | Catalog #16210-072 |
| Normal donkey serum | Millipore Sigma | Catalog #S30-M |
| **Software** | | |
| Fiji | Open Source | RRID:SCR_002285; http://fiji.sc |
| GraphPad Prism | Licensed Software | RRID:SCR_002798; http://www.graphpad.com |
| CellProfiler Image Analysis Software | Open Source | RRID:SCR_007358; http://cellprofiler.org |
| ChEA3 | Open Source | RRID:SCR_005403; http://amp.pharm.mssm.edu/lib/chea.jsp |
| JASPAR | Open Source | RRID:SCR_003030; http://jaspar.genereg.net |
| PROMO | Open Source | RRID:SCR_016926; http://alggen.lsi.upc.es/cgi-bin/promo_v3/promo/promoinit.cgi?dirDB=TF_8.3 |
| Imaris | Licensed Software | RRID:SCR_007370; http://www.bitplane.com/imaris |

| Reagent/resource | Reference or source | Identifier or catalog number |
|---|---|---|
| **Others** | | |
| Dual-Luciferase Reporter Assay System | Promega | Catalog #E1910 |
| Click-iT EdU Cell Proliferation Kit for Imaging, Alexa Fluor 647 dye | Thermo Fisher Scientific | Catalog #C10340 |
| Direct-zol RNA Miniprep Kits | Zymo Research | Catalog #R2050 |
| TruSeq RNA Library Prep Kit | Illumina | Catalog #RS-122-2001 |
| Ribo-Zero rRNA Removal Kit | Epicentre | Catalog #RZH1086 |
| CUTANA ChIC/ CUT&RUN kit | EpiCypher | Catalog #14-1048 |
| Duolink In Situ Red Starter Kit Mouse/Rabbit | Millipore Sigma | Catalog #DUO92101 |
| Dual-Luciferase Reporter Assay System | Promega | Catalog #E1910 |

## Mice

Animal treatment and care followed NIH guidelines and the requirements of Carnegie Institution, and approved by Carnegie Institutional Animal Care and Use Committee. $Mpp7^{flox}$ mice was generated via contractual service with ALSTEM Inc., and available upon request. $Pax7^{CreERT2}$ mice (Lepper et al, 2009) were donated by Dr. C Lepper. $Amot^{flox}$ mice was obtained from Dr. J Kissil (Shimono and Behringer, 2003). $Yap^{flox}$, $Taz^{flox}$, and $Rosa26^{YFP}$ mice (Reginensi et al, 2013; Srinivas et al, 2001) were obtained from The Jackson Laboratory. Mice were genotyped by PCR using tail DNA by allele-specific oligonucleotides (information available upon request). Appropriate mating schemes were performed to obtain control and experimental mice stated in text, figures and legends. Both male and female mice were used and included in data analysis unless specified otherwise.

## Animal procedures

Mice (3–6 month of age) were administered intraperitoneally for 5 consecutive days with 250 L tamoxifen (10 mg/mL corn oil; Millipore Sigma), followed by 3 days of chase. For muscle injury, mice were anesthetized and 50 µL of 10 µM cardiotoxin (CTX; Millipore Sigma) in PBS was injected into tibialis anterior (TA) muscles using the BD insulin syringe (Becton Dickenson). For EdU (5-ethynyl-2'-deoxyuridine; Millipore Sigma) incorporation in vivo, 10 µL of EdU (0.5 mg/ml in PBS) per gram of weight was used per intraperitoneal injection. Time lines of experimental procedure and muscle sample harvest are detailed in figures and legends.

## Histology and immunofluorescence (IF)

TA muscles were fixed in 4% PFA (Electron Microscopy Sciences) immediately after harvesting. They were processed through 10%

sucrose/PBS, 20% sucrose/PBS, and FSC 22 frozen section media (Leica) before mounted onto a cork and flash-frozen in liquid nitrogen-cooled isopentane (VWR). Frozen samples were stored in −80 °C until sectioning by a cryostat (Leica CM3050 S). Sections of 10 μm thickness were collected on Superfrost plus slides (VWR), dried, and stored at −20 °C for future use. For histology, Hematoxilin Gill's II and Eosin (H&E) were used following instructions of the manufacturer (Surgiopath), and mounted in Permount (VWR). For IF, sections were permeabilized with 0.6% TritonX-100/PBS for 20 min, blocked in Mouse on Mouse (M.O.M; Vector) blocking reagent, and then in blocking buffer (10% normal goat serum (Gibco) or normal donkey serum (Sigma-Aldrich), 10% carbo-free blocking solution (Vector) in PBS). Harvested single myofibers and cultured cells (see below) were fixed and permeabilized the same way and blocked in blocking buffer without M.O.M. reagent. Tissues and cells were incubated with primary antibodies overnight at 4 °C and secondary antibodies for 1 h at room temperature. DAPI was used to detect nuclei. EdU incorporation was detected by Click-iT Alexa Fluor 647 Imaging kit (Thermo Fisher Scientific). Brightfield microscope (Nikon Eclipse E 800), fluorescence microscope (Nikon E800), and confocal microscope (Leica TCS SP5) were used for imaging. For fluorescent signal intensities and authentication of antibodies, fluorescent secondary antibody alone was used as a reference baseline (example of anti-MPP7 IF in Appendix Fig. S9).

## MuSC isolation by fluorescence-activated cell sorter (FACS)

YFP-labeled MuSCs were isolated by FACS from control and cKO mice specified in text, figures, and legends. Briefly, hindlimb muscles were minced and digested with 0.2% collagenase (Worthington Biochemical) for 90 min followed by 0.2% dispase (Thermo Fisher Scientific) for 30 min in 37 °C shaking water bath. Triturated muscle suspension was filtered through a 40 μm cell strainer (Corning) and subjected to isolation by BD FACSAriaIII. For culture, mononuclear cells were seeded on Matrigel (Corning) coated plates in growth media (DMEM with 20% FBS, 5% horse serum, 1% pen-strep, 1% glutamax (above from Gibco), 0.1% chick embryo extract (MPbio) and 2 ng/mL FGF2; R&D systems) for specified time in text and legends, before fixation and analysis.

## Single myofiber isolation

Single myofibers were isolated from extensor digitorum longus (EDL) muscles as described (Li and Fan, 2017). Isolated MuSCs were fixed in 4% PFA immediately or cultured in DMEM with 10% horse serum and 0.5% chick embryo extract for specified time in text, figures, and legends. To preserve MuSC projections, modifications were made to the procedure (Kann et al, 2022). Knee tendon was cut prior to ankle tendon to remove the EDL muscle. Tugging and pulling were avoided to prevent muscle stretching and loss of projections. EDL muscles were digested in 2.6 mg/mL collagenase in DMEM with Y-27632 (50 μM; Tocris Bioscience) for 55 min in 37 °C shaking water bath, and transferred to DMEM with Y-27632 (50 μM) for trituration to liberate individual myofibers. These single myofibers were immediately fixed in 4% PFA. After fixation, myofibers and their associated MuSCs were subjected IF and imaging analysis.

## PLA assay

Single myofibers were isolated and fixed as described above, and then processed using the Duolink PLA fluorescence kit (mouse and rabbit antibody combination, 568 nm detection; Millipore Sigma). Briefly, single myofibers were permeabilized with 0.6% Triton X-100 for 20 min and blocked with PLA blocking solution and incubated with primary antibodies overnight. The concentrations of primary antibodies were doubled than normal IF staining. Then the PLA probes were added and ligated followed by signal amplification. Myofibers were mounted with ProLong Diamond anti-fade mountant (Invitrogen) for confocal imaging. The Imaris software was used for 3D reconstruction and quantifying the PLA dots (see below).

## RNA-seq and analyses

For RNA-seq, 3-month-old female mice were used. YFP-labeled MuSCs were purified by FACS and processed for RNA extraction using Direct-zol RNA Miniprep Kit (Zymo Research). Total RNA was processed by ribosomal RNA depletion using the Ribo-Zero rRNA Removal Kit (Illumina) and sequencing library generated using the TruSeq RNA Library Prep Kit (Illumina) with omission of PolyA selection. Raw data from FastQ were processed using standard method (Pertea et al, 2016) and the reads were mapped to the mouse mm9 genome. Differentially expressed gene (DEG) analysis was performed with DESeq2 (Love et al, 2014) with default parameters. Transcription factors binding sites in gene promotors were identified by using ChEA3 (Keenan et al, 2019) and GeneHancer prediction (Fishilevich et al, 2017). To cross-compare our DEGs were with *Yap/Taz* overexpression (Sun et al, 2017) and *Yy1* cKO (Chen et al, 2019), respectively for analyses. Significance of the rate of enrichment was assessed using hypergeometric test, and *P*-values stipulated in figures.

## Plasmid transfection of single myofibers and 293T cells

For single myofiber transfection, myofibers isolated from Con, *Mpp7* cKO or *Amot* cKO were cultured for 12 h and transfected with indicated expression plasmids using TransfeX reagent (ATCC). Myofibers were then cultured and harvested at indicated time; 1 μM 4-OH-TMX (Tocris Bioscience) was added to sustain knockout efficiency. For 293T cells, indicated plasmids were transfected using Lipofectamine 3000 (Thermo Fisher Scientific). After 24 hr, cells were lysed by RIPA buffer (50 mM Tris-HCl [pH 8.0], 150 mM NaCl, 1% NP-40, 0.5% deoxycholic acid, and 0.1% SDS) supplemented with complete protease inhibitor cocktail (Roche) and 1 mM PMSF (Millipore Sigma) processed for western blot detection.

## Co-immunoprecipitation (co-IP) and western blot

Cell lysates from transfected 293T were incubated with anti-FLAG M2 magnetic beads (Millipore Sigma) or anti-V5 agarose beads (Millipore Sigma) at 4 °C for 4 h or overnight. Beads were then washed three times with NP-40 cell lysis buffer (50 mM Tris-HCl [pH 8.0], 150 mM NaCl, 1% NP-40 supplemented with 0.5 mM dithiothreitol) and one time with PBS. 5% input and immunoprecipitated fractions were boiled in Laemmli SDS sample buffer

(Thermo). Protein samples were processed for SDS-PAGE (4–15% gel; Bio-Rad), transferred to PDMF membrane (Bio-Rad) for detection using rabbit anti-HA, rabbit anti-V5, or rabbit anti-FLAG antibodies (Cell Signaling), followed by HRP-conjugated goat anti-rabbit antibodies (Bio-Rad). ECL substrate (Thermo Fisher Scientific) was used for detection. Exposure and images were performed using LI-COR Fc imager (LI-COR Biosciences).

## Luciferase assay

Carm1 promoter region ($-630$ to $+15$) was cloned to pGL4 luciferase reporter vector (Promega) to be the Carm1-reporter. Carm1 promoter region was analyzed by PROMO (Farre et al, 2003; Messeguer et al, 2002) and JASPAR (Castro-Mondragon et al, 2022) to identify TEAD and YY1 binding sites. Carm1-reporter with TEAD and YY1 binding site mutated were generated by PCR and the nucleotide sequences are indicated in figures. The Carm1-reporter or its mutated reporters was co-transfected into 293T with the pRL-TK plasmid (Promega) expressing renilla for normalization. Combinations of cDNA expression plasmids were indicated in figures and legends. Twenty-four h after transfection, 293T cells were harvested and luciferase and renilla activities were detected using the Dual-luciferase reporter assay kit (Promega) in a Glowmax 20/20 luminometer (Promega). The 8XGTIIC-luciferase vector (Dupont et al, 2011) was used as the TEAD-reporter using the same procedure. For reporter assays, three independent biological replicates were performed for each combination of reporters and cDNA expression plasmids.

## Pharmacological treatments

For EdU incorporation in cultured MuSCs, EdU was mixed in media to a final concentration of $10\,\mu M$ for 24 hr, followed by the Click-reaction (Thermo Fisher Scientific) for detection. FACS-isolated MuSCs were immediately treated with DMSO (mock-treatment), Jasplakinolide (100 nM; Tocris Bioscience) or Narciclasine (100 nM; Tocris Bioscience) at for 2 h, cytospun to coverslip, and processed for IF. For activated MuSCs, they were cultured for 48 h after FACS-isolation, and treated by DMSO, Blebbinstatin ($10\,\mu M$; Tocris Bioscience), Cytochalasin B ($10\,\mu M$; Tocris Bioscience), and Y-27632 ($10\,\mu M$) for 2 h and processed for IF.

## ChIP-qPCR

In total, $5 \times 10^5$ MuSC-derived myoblasts cultured in GM for 2 days were processed by using the ChIC/CUT&RUN kit (EpiCypher) following manufacturer's protocol. Briefly, cells were fixed for 5 min in 0.1% formaldehyde and immobilized on ConA beads. Then 2 μg of each antibody (anti-rabbit IgG, anti-MPP7, anti-TAZ, and anti-YY1) were added and incubated overnight on nutator. Cells were then permeabilized by 0.05% Digitonin and digested in pAG-MNase/Calcium for 2 h. Digested chromatin was reverse cross-linked at 55 °C and subjected to DNA purification and quantification (Q-bit). For qPCR samples, 50 ng DNA was used with SsoAdvanced Universal SYBR Green Supermix (Qiagen) in the Bio-Rad CFX96-Real Time System for data collection and quantification using the Bio-Rad CFX Maestro software package. Primer sequences are shown in reagents and tools table.

## Quantifications and statistical analysis

For cryosections, ≥50 cells from 10 sections per animal (5 animals per genotype) were imaged using a Nikon E800 fluorescence microscope at ×40 magnification. For MuSC fractions on single myofibers, total 150–200 cells were assessed from 2 to 3 animals using the same microscopy above. For quantification of IF signal intensity, 50 cells on myofibers (from 2 to 3 mice) or 200 dish-cultured MuSCs (from 2 to 3 mice) were imaged at 63×/1.4 oil fluorescent objectives on a Leica TCS SP5 confocal microscope. Gain and exposure settings were consistent between experiments. Z-stacks were collected to capture full objection depth of MuSCs. Images were exported to Fiji and CellProfiler for analysis. For images without 3D presentation, single mid-plane images were used to best represent sub-cellular distribution, and the conclusion was based on Z-stacks. For 3D reconstruction and quantitative analysis, Imaris was used.

For quantification of IF signal intensity and nuclear vs. cytoplasmic distribution, single myofibers or cultured cells were IF-stained for protein of interest (i.e., MPP7, AMOT, YAP, TAZ, CARM1, or tagged epitopes), YFP and DAPI, and multi-channel images were acquired. A custom CellProfiler pipeline was used to set threshold on YFP and DAPI channels to identify primary and secondary objects, respectively. The primary objects from YFP channel were then used as masks on protein-of-interest channel images, and the integrated intensity of the masked image was used for total signal intensity. The secondary objects from DAPI channel were used as masks on protein-of-interest channel images and the intensity of the masked part was quantified for nuclear signal intensity. For antibody validation and background threshold, spectra distribution from stained cell populations were plotted to determine positive and negative signal cutoff value (Appendix Fig. S9).

For co-IP quantification, the samples were analyzed in three independent biological replicates. Intensity of blotting bands were measured using Fiji. Co-IPed target proteins were then normalized to their primary IP proteins.

For statistics, error bars represent means ± SD. Data analyses were performed by Prism 9 software. Data comparison of two independent groups was performed by two-tailed unpaired Student's $t$ test. Multiple group analysis was performed using one-way ANOVA followed by the Tukey or Dunnet post hoc test for multiple comparison per figure legends. To test significance of cell population fraction, the total SC population over all the experimental repeats were included and comparisons were performed by Chi-squared test. ***$P \leq 0.001$; **$P \leq 0.01$; *$P \leq 0.05$; n.s, not significant, $P > 0.05$. For RNA-seq, $Padj < 0.05$ was considered significant. To calculate the statistical significance of overlapped genes in Venn diagrams, $P$ values were calculated based on hypergeometric test.

## Data availability

RNA-seq data in this study have been deposited to Gene Expression Omnibus (GEO) database under the accession GSE241340.

The source data of this paper are collected in the following database record: biostudies:S-SCDT-10_1038-S44319-024-00305-4.

## Peer review information

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

## Acknowledgements

We thank Dr. Lydia Li for the initial finding. Dr. Wanjin Hong for sharing the Taz (WT and WWm) plasmids. Dr. Frederick Tan for training in transcriptome analysis, Allison Pinder for assistance in RNA-seq, Dr. Mahmud Siddiqui for assistance in microscopy, Drs. Liangji Li and Haolong Zhu for assistance with FACS and single myofiber culture, and Colin Bylieu for tail PCR genotyping. We also thank the Carnegie rodent facility crew. C-MF was supported by the NIH (R01AR060042, R01AR071976, and R01AR072644) and the Carnegie Institution for Science. AS was supported by the NIH (R01AR072644). JK is supported by the NIH (5R01NS117926 and R33NS119658).

## Author contributions

**Anwen Shao**: Conceptualization; Data curation; Formal analysis; Investigation; Methodology; Writing—original draft; Writing—review and editing. **Joseph L Kissil**: Resources; Writing—review and editing. **Chen-Ming Fan**: Conceptualization; Resources; Data curation; Software; Formal analysis; Supervision; Funding acquisition; Writing—original draft; Project administration.

Source data underlying figure panels in this paper may have individual authorship assigned. Where available, figure panel/source data authorship is listed in the following database record: biostudies:S-SCDT-10_1038-S44319-024-00305-4.

## Disclosure and competing interests statement

The authors declare no competing interests.

