## [Peer Review File · EMBO Reports]

The L27 Domain of MPP7 enhances TAZ-YY1 Cooperation to Renew Muscle Stem Cells

Anwen Shao, Joseph Kissil, and Chen-Ming Fan

Corresponding author(s): Chen-Ming Fan (fan@carnegiescience.edu)

Review Timeline:

Submission Date:	4th Jun 24
Editorial Decision:	11th Jul 24
Revision Received:	26th Aug 24
Editorial Decision:	2nd Oct 24
Revision Received:	17th Oct 24
Accepted:	22nd Oct 24

Editor: Achim Breiling

Transaction Report:

Dear Dr. Fan,

Thank you for the submission of your manuscript to EMBO reports. I have received the reports from two the three referees that were asked to evaluate your study, which can be found at the end of this email. A third referee agreed to evaluate the study, but never submitted a report, despite several reminders from our side. I thus decided to proceed with the two reports I have.

As you will see, the referees have several comments, concerns, and suggestions, indicating that a major revision of the manuscript is necessary to allow publication of the study in EMBO reports. As the reports are below, and all the concerns need to be addressed, I will not detail them further here.

Given the constructive referee comments, I would like to invite you to revise your manuscript with the understanding that the concerns of the referees must be addressed in the revised manuscript and in a detailed point-by-point response. Acceptance of your manuscript will depend on a positive outcome of a second round of review. It is EMBO reports policy to allow a single round of revision only and acceptance of the manuscript will therefore depend on the completeness of your responses included in the next, final version of the manuscript.

- 1) a .docx formatted version of the final manuscript text (including legends for main figures, EV figures and tables), but without the figures included. Figure legends should be compiled at the end of the manuscript text.
- 2) individual production quality figure files as .eps, .tif, .jpg (one file per figure), of main figures and EV figures. Please upload these as separate, individual files upon re-submission.

- 4) a complete author checklist, which you can download from our author guidelines

(<https://www.embopress.org/page/journal/14693178/authorguide>). Please insert page numbers in the checklist to indicate where the requested information can be found in the manuscript. The completed author checklist will also be part of the RPF.

5) that primary datasets produced in this study (e.g. RNA-seq, ChIP-seq, structural and array data) are deposited in an appropriate public database. If no primary datasets have been deposited, please also state this in a dedicated section (e.g. 'No primary datasets have been generated and deposited'), see below.

The accession numbers and database should be listed in a formal "Data Availability" section (placed after Materials & Methods) that follows the model below. This is now mandatory (like the COI statement). Please note that the Data Availability Section is restricted to new primary data that are part of this study. This section is mandatory. As indicated above, if no primary datasets have been deposited, please state this in this section

Data availability

8) Regarding data quantification and statistics, please make sure that the number "n" for how many independent experiments were performed, their nature (biological versus technical replicates), the bars and error bars (e.g. SEM, SD) and the test used to calculate p-values is indicated in the respective figure legends (also for EV figures and all those in an Appendix). Please also check that all the p-values are explained in the legend, and that these fit to those shown in the figure. Please provide statistical testing where applicable. Please avoid the phrase 'independent experiment', but clearly state if these were biological or technical replicates. Please also indicate (e.g. with n.s.) if testing was performed, but the differences are not significant. In case n=2, please show the data as separate datapoints without error bars and statistics. See also: <http://www.embopress.org/page/journal/14693178/authorguide#statisticalanalysis>

9) Please add scale bars of similar style and thickness to microscopic images, using clearly visible black or white bars (depending on the background). Please place these in the lower right corner of the images themselves. Please do not write on or near the bars in the image but define the size in the respective figure legend.

10) Please also note our reference format:

12) We now use CRediT to specify the contributions of each author in the journal submission system. CRediT replaces the author contribution section. Please use the free text box to provide more detailed descriptions and do NOT provide your final manuscript text file with an author contributions section. See also our guide to authors: <https://www.embopress.org/page/journal/14693178/authorguide#authorshipguidelines>

13) All Materials and Methods need to be described in the main text using our 'Structured Methods' format, which is required for all research articles. According to this format, the Materials and Methods section should include a Reagents and Tools Table (listing key reagents, experimental models, software, and relevant equipment and including their sources and relevant identifiers), uploaded as separate file, followed by a Methods and Protocols section in which we encourage the authors to describe their methods using a step-by-step protocol format with bullet points, to facilitate the adoption of the methodologies across labs. More information on how to adhere to this format as well as downloadable templates (.doc) for the Reagents and Tools Table can be found in our author guidelines (section 'Structured Methods'):

14) Please add 5 keywords to the manuscript text file and order the manuscript sections like this, using these names: Title page - Abstract - Keywords - Introduction - Results - Discussion - Methods - Data availability section - Acknowledgements - Disclosure and Competing Interests Statement - References - Figure legends - Expanded View Figure legends

I look forward to seeing a revised version of your manuscript when it is ready. Please let me know if you have questions or comments regarding the revision.

Yours sincerely,

Referee #1:

First, Shao and colleagues demonstrated that MuSC-specific Mpp7 conditional knockout (KO) exhibited impaired regeneration, reduced proliferative capacity, and a decreased number of self-renewed MuSCs. The authors identified the L27-domain of Mpp7 as involved in self-renewal of MuSCs through a rescue experiment using SM. Based on previous data showing the binding of Mpp7 and Amot, the authors focused on interaction between Mpp7 and Amot, finding that Mpp7 and Amot bind to each other via their respective PDZ and PDM domains. Furthermore, the authors showed a similar regenerative defect in Amot-cKO as in Mpp7-cKO.

Next, the authors investigated the genes commonly altered in Mpp7- and Amot-cKO MuSCs and identified Carm1, which methylates Pax7. Interestingly, the authors demonstrated that the forced expression of Carm1 was sufficient to rescue the defect of Mpp7 in MuSCs. During these analyses, the authors noted the presence of YY1 binding sites in the promoters of common target genes for Mpp7 and YAP/TAZ, in addition to TEAD binding sites. They found that: 1) the TEAD binding site is required for Carm1 expression, and Mpp7 and Amot significantly enhance its expression level, and 2) in the active form of Taz, the proportion of Pax7+ Myod+ cells could be rescued in the absence of Mpp7, but not self-renewal. Furthermore, using several mutated proteins, the authors demonstrated 1) the relationship between F-actin and AMOT localization, and 2) that MPP7, AMOT, TAZ, and YY1 form a robust transcription activator complex, essential for Carm1 expression.

Overall, the experiments were well-conducted, and the data were convincing. Meanwhile, there is no conclusive evidence that the formation of this complex is necessary for the self-renewal of MuSCs. The data on self-renewal in vivo could be a secondary effect, as myofiber formation was impaired. The in vitro SM experiment was one experimental system to verify the self-renewal of MuSCs. However, the work is sufficiently novel and interesting as a regulatory mechanism for the expression of critical genes in MuSCs. Therefore, this reviewer recommends publishing this work in EMBO Reports.

Specific comments

1) Comment; During the muscle regeneration process, the proliferation of MuSCs has two phases. The first is the proliferation that peaks on the third day of regeneration, and the second is around the fifth day of regeneration. According to the work of

Olwin's group, self-renewal of MuSCs occurs around day 5. If L27 of Mpp7 has a limited role in the self-renewal of MuSCs, is the complex identified in this study characteristically present in Pax7-positive cells around day 5 of regeneration? As mentioned below, Carm1 is also expressed in differentiating muscle cells. It would therefore be useful for readers if the authors could explain how the complex discovered here functions in self-renewing MuSCs.

[https://www.cell.com/science/fulltext/S2589-0042\(22\)00715-5](https://www.cell.com/science/fulltext/S2589-0042(22)00715-5)

2) Fig. 2C

What is the possible mechanism by which Δ L27-Mpp7 restored the proportion of Pax7+MyoD+ cells? Does this indicate that the nuclear translocation of YAP shown in previous studies by the authors occurs normally with Δ L27-Mpp7? Could this be confirmed by staining for YAP/TAZ?

3) Fig. 3H

The following study showed that Carm1 was involved in myogenic differentiation. In contrast, current data demonstrated that Carm1 functions as an anti-myogenic factor, as the number of committed myogenic cells (MyoD+Pax7-, perhaps Myogenin+) was reduced. If possible, this reviewer recommends discussing the role of Carm1 in myogenic differentiation processes.

J Biol Chem. 2002 Feb 8;277(6):4324-33. doi: 10.1074/jbc.M109835200

4) Lines 225-229; 'Using TAZ as a representative (for TAZ and YAP), we found that Mpp7 co-expression indeed increased Taz's transcriptional activity on a TEAD-reporter³⁶ (Fig. 4e). Amot alone inhibited Taz, but Amot, Mpp7, and Taz altogether best activated the reporter. Co-IP also revealed an enhanced interaction between MPP7 and TAZ by AMOT (Fig. 4f; Supplementary Fig. S4e).

Comment; The explanations for Fig 4e and Fig 4f are switched, so please correct them.

5)Fig. S5d

Comment; Is there really a significant difference between V5-Taz and V5-Tax S89A?

6)In Fig. 6e

Comment; There is a lack of explanation for the result indicating that S175A Amot does not affect self-renewal.

7)Lines 292-293; As such, AMOT is likely the F-actin-responsive conduit for nuclear MPP7 and YAZ/TAZ for MuSC renewal. Comment; As the authors described, MuSC activation is accompanied by actin rearrangement. However, to the reviewer's knowledge, it is unclear whether the actin rearrangement is required for the self-renewal of MuSCs. If possible, the reviewer recommends adding evidence indicating the necessity of actin rearrangement for the self-renewal of adult stem cells.

Referee #2:

In this manuscript the authors describe a new function of MPP7, AMOT, and to some extent YAP/TAZ in the regeneration of muscle stem cells after injury. This follows the description, in 2017, of the MPP7-AMOT axis. In this manuscript, the authors advocate for the existence of a MPP7-AMOT-TAZ transcriptional complex, for which they provide evidence at the level of protein-protein interactions, cooperation on gene expression (RNAseq and luciferase assays), and similarity of in vivo phenotypes. This complex is not a general regulator of YAP/TAZ function, but only affects a subset of genes, likely because these genes are also regulated by the YY1 transcription factor, binding in the vicinity of TAD/TAZ complexes.

Overall the study looks pretty advanced. I have some remaining concerns:

1) a main proposed model of the manuscript is that AMOT needs to be incorporated into the MPP7-TAZ complex for it to work efficiently. If so, a WW-mutant of TAZ should be functionally inactive, both on luciferase assays and on MUSC differentiation assays. This is important because in the past many claimed that the WW domain mediates important functions of YAP and/or TAZ based on biochemistry data (protein-protein interaction), without otherwise providing more important functional data.

2) the authors focused on TAZ, but TAZ (at difference with CARM1) GOF is very weak in the MUSC differentiation assay. This could be due to some degree of functional redundancy with YAP. The authors should try YAP in their functional assays (see point 1), also in light of the strong phenotype of YAP/TAZ DKO.

3) The axis with CARM1 is proposed as a main mediator of the end point muscle phenotype. If so, could the authors use the MUSC differentiation assay to probe whether CARM1 GOF also rescue, at least to some extent, YAP/TAZ KO MUSCS? This would be important not to validate/invalidate the model, but rather to gauge its overall strength and importance in the context of the wider biology.

Dear Editor,

We thank you and the reviewers for your time in processing our manuscript. We also very much appreciate the reviewers' comments. Please see our point-by-point response to reviewers' comments below. For easy identification of contents, reviewers' comments are *italicized*.

Referee #1:

First, Shao and colleagues demonstrated that MuSC-specific Mpp7 conditional knockout (KO) exhibited impaired regeneration, reduced proliferative capacity, and a decreased number of self-renewed MuSCs. The authors identified the L27-domain of Mpp7 as involved in self-renewal of MuSCs through a rescue experiment using SM. Based on previous data showing the binding of Mpp7 and Amot, the authors focused on interaction between Mpp7 and Amot, finding that Mpp7 and Amot bind to each other via their respective PDZ and PDM domains. Furthermore, the authors showed a similar regenerative defect in Amot-cKO as in Mpp7-cKO.

Next, the authors investigated the genes commonly altered in Mpp7- and Amot-cKO MuSCs and identified Carm1, which methylates Pax7. Interestingly, the authors demonstrated that the forced expression of Carm1 was sufficient to rescue the defect of Mpp7 in MuSCs. During these analyses, the authors noted the presence of YY1 binding sites in the promoters of common target genes for Mpp7 and YAP/TAZ, in addition to TEAD binding sites. They found that: 1) the TEAD binding site is required for Carm1 expression, and Mpp7 and Amot significantly enhance its expression level, and 2) in the active form of Taz, the proportion of Pax7+ Myod+ cells could be rescued in the absence of Mpp7, but not self-renewal. Furthermore, using several mutated proteins, the authors demonstrated 1) the relationship between F-actin and AMOT localization, and 2) that MPP7, AMOT, TAZ, and YY1 form a robust transcription activator complex, essential for Carm1 expression.

Overall, the experiments were well-conducted, and the data were convincing.

Meanwhile, there is no conclusive evidence that the formation of this complex is necessary for the self-renewal of MuSCs. The data on self-renewal in vivo could be a secondary effect, as myofiber formation was impaired. The in vitro SM experiment was one experimental system to verify the self-renewal of MuSCs. However, the work is sufficiently novel and interesting as a regulatory mechanism for the expression of critical genes in MuSCs. Therefore, this reviewer recommends publishing this work in EMBO Reports.

We thank reviewer 1 for the positive recommendation for publishing our manuscript.

Specific comments

1) Comment; During the muscle regeneration process, the proliferation of MuSCs has two phases. The first is the proliferation that peaks on the third day of regeneration, and the second is around the fifth day of regeneration. According to the work of Olwin's group, self-renewal of MuSCs occurs around day 5. If L27 of Mpp7 has a limited role in

the self-renewal of MuSCs, is the complex identified in this study characteristically present in Pax7-positive cells around day 5 of regeneration? As mentioned below, Carm1 is also expressed in differentiating muscle cells. It would therefore be useful for readers if the authors could explain how the complex discovered here functions in self-renewing MuSCs.

[https://www.cell.com/science/fulltext/S2589-0042\(22\)00715-5](https://www.cell.com/science/fulltext/S2589-0042(22)00715-5)

Thank you for this comment. We have now included additional comments on the timing of renewed quiescent MuSC relevant to our work (ln 123-127); the above reference is also added:

“The majority of renewed quiescent MuSCs are documented to be derived from cell divisions at d 5 and onwards after injury (Cutler et al., 2022). Our data therefore suggest either that the early proliferation defect (1-5 dpi) of *Mpp7* cKO leads to the reduction of renewed quiescent SCs at 21 dpi or that *Mpp7* also acts in later renewal divisions prior to quiescence, and possibly both.”

Briefly, the work by Cutler et al. (2022) provided data that after injury, MuSCs that divide (assessed by EdU incorporation) in the time window of d 5-7 contributed to the largest fraction of MuSCs that retained EdU and returned to quiescence at d14. MuSCs divided in d 0-4 contributed to a smaller fraction (~1/2 of those in d 5-7) of SCs returned to quiescence. This was based on MuSCs incorporated EdU at select time windows and still retained detectable level of EdU at d 14. Our understanding of their main conclusion is that more ‘return-to-quiescence’ MuSCs undergo their last round(s) of division from d 5 and onwards and retain sufficient EdU for detection. This does not exclude the possibility that many MuSCs divide at d2-4 continue to divide and dilute EdU to below detection later in quiescence. An additional complication is that we do not know, in each select time window, the fraction of MuSCs that undergo parental-DNA-strand-retention-division and are not labeled by EdU. In our view, all regenerative MuSCs are derived from the original pool of Pax7+ MuSCs prior to injury. Therefore, defects in earlier divisions likely lead to a smaller pool of SCs that can later divide and return to quiescence. Our data do show that an earlier defect in EdU incorporation and Pax7 cell number at d 5 (of *Mpp7* cKO) and the reduction of return-to-quiescence Pax7+ MuSCs at d 21. Given that all components in our proposed protein complex are pleiotropically expressed, they are likely ready for action at all times unless prevented by other regulatory mechanisms (e.g., F-actin state/cytoplasmic retention).

Importantly, we consider that renewal divisions do not have to be those just prior to returning to quiescence, but also include earlier divisions to maintain the pool of SCs that later continue to divide and return to quiescence. The original work on Carm1’s role in MuSC renewal also focused on symmetric vs asymmetric renewal divisions (instead of returning to quiescence). Regarding Carm1’s role in myogenic differentiation, please see our response to comment 3 below.

2) Fig. 2C

What is the possible mechanism by which Δ L27-Mpp7 restored the proportion of

Pax7+MyoD+ cells? Does this indicate that the nuclear translocation of YAP shown in previous studies by the authors occurs normally with Δ L27-Mpp7? Could this be confirmed by staining for YAP/TAZ?

We have now addressed this question with new data. As expected, Δ L27-Mpp7 restored the level of TAZ/YAP, explaining why it restores the progenitor fate (as TAZ or YAP does alone in this context). The data is now included as Fig.S4h with accompanying text (ln 265-270) and legend.

We note that the text flow does not allow us to place the data immediately following the Mpp7 domain deletion/renewal data (Fig. 2), as the TAZ/YAP connection is introduced later (Fig. 4). As such, we placed this data in Fig.S4h, albeit slightly meandering in the main text.

3) Fig. 3H

The following study showed that Carm1 was involved in myogenic differentiation. In contrast, current data demonstrated that Carm1 functions as an anti-myogenic factor, as the number of committed myogenic cells (MyoD+Pax7-, perhaps Myogenin+) was reduced. If possible, this reviewer recommends discussing the role of Carm1 in myogenic differentiation processes.

J Biol Chem. 2002 Feb 8;277(6):4324-33. doi: 10.1074/jbc.M109835200

Thank you for bringing this to our attention. We agree and have now included a statement (and the reference) for Carm1's role in myogenic differentiation in the discussion (ln 385-385):

"Worth noting is that CARM1 has also been documented to potentiate myogenic differentiation via distinct interacting proteins."

4) Lines 225-229; 'Using TAZ as a representative (for TAZ and YAP), we found that Mpp7 co-expression indeed increased Taz's transcriptional activity on a TEAD-reporter36 (Fig. 4e). Amot alone inhibited Taz, but Amot, Mpp7, and Taz altogether best activated the reporter. Co-IP also revealed an enhanced interaction between MPP7 and TAZ by AMOT (Fig. 4f; Supplementary Fig. S4e).

Comment; The explanations for Fig 4e and Fig 4f are switched, so please correct them.

Thank you for pointing this out. We have corrected the error by re-phrasing those sentences (now ln. 230-232):

"...we found an enhanced interaction between MPP7 and TAZ by AMOT using co-IP assay (Fig. 4e; Supplementary Fig. S4i). Furthermore, Mpp7 co-expression indeed increased Taz's transcriptional activity on a TEAD-reporter (Fig. 4f)."

5)Fig. S5d

Comment; Is there really a significant difference between V5-Taz and V5-Tax S89A?

Apologies, we accidentally put the wrong chart in the previous subfigure (Fig. S5d), which was a duplicate of Fig. 4h. It should have been the quantification for Carm1 levels by expressing L27-Taz vs Taz, and this has been corrected (currently S5e). Regarding Fig. 4h, although it does not look like there is a difference, the p-value does indicate a significant difference between Taz and Taz S89A.

6) *In Fig. 6e*

Comment; There is a lack of explanation for the result indicating that S175A Amot does not affect self-renewal.

We have now added an explanation (ln 304-305):

“S175A Amot has been shown to elevate nuclear YAP, which helps explain the rescue of progenitor fate.”

7) *Lines 292-293; As such, AMOT is likely the F-actin-responsive conduit for nuclear MPP7 and YAP/TAZ for MuSC renewal.*

Comment: As the authors described, MuSC activation is accompanied by actin rearrangement. However, to the reviewer's knowledge, it is unclear whether the actin rearrangement is required for the self-renewal of MuSCs. If possible, the reviewer recommends adding evidence indicating the necessity of actin rearrangement for the self-renewal of adult stem cells.

Thank you for pointing out the lack of direct evidence to support actin re-arrangement being a requirement for renewal to date. We used ‘likely’ for a speculative proposal but understand the reviewer’s concern. We have now deleted this sentence, as the prior sentence interprets/summarizes the data without the need to speculate further.

Referee #2:

In this manuscript the authors describe a new function of MPP7, AMOT, and to some extent YAP/TAZ in the regeneration of muscle stem cells after injury. This follows the description, in 2017, of the MPP7-AMOT axis. In this manuscript, the authors advocate for the existence of a MPP7-AMOT-TAZ transcriptional complex, for which they provide evidence at the level of protein-protein interactions, cooperation on gene expression (RNAseq and luciferase assays), and similarity of in vivo phenotypes. This complex is not a general regulator of YAP/TAZ function, but only affects a subset of genes, likely because these genes are also regulated by the YY1 transcription factor, binding in the vicinity of TAD/TAZ complexes.

Overall the study looks pretty advanced. I have some remaining concerns:

1) a main proposed model of the manuscript is that AMOT needs to be incorporated into

the MPP7-TAZ complex for it to work efficiently. If so, a WW-mutant of TAZ should be functionally inactive, both on luciferase assays and on MUSC differentiation assays. This is important because in the past many claimed that the WW domain mediates important functions of YAP and/or TAZ based on biochemistry data (protein-protein interaction), without otherwise providing more important functional data.

Following the reviewer's comment, we have now included:

1) By luc assay, Taz-WWm (has its own TAD) could activate the TEAD-reporter, but not to the same level as wt Taz (new Fig. S5c). This new figure is placed next to the co-IP data showing a lack of interaction between Mpp7 and Taz-WWm. The reduced activity is likely due to its inability to interact with endogenous Amot and Mpp7 for higher transcriptional activity as wt Taz in 293T cells.

2) By SM assay (new Fig 4i), Taz-WWm is also not able to rescue Mpp7 cKO renewal. Due to the text flow and presenting the data next to each other for comparison, we merge this data with that of wt Taz. As such, we refer to it after the Luc data. It is a bit unconventional, but the best option.

The accompanying description in the main text is in ln 260-262.

2) the authors focused on TAZ, but TAZ (at difference with CARM1) GOF is very weak in the MUSC differentiation assay. This could be due to some degree of functional redundancy with YAP. The authors should try YAP in their functional assays (see point 1), also in light of the strong phenotype of YAP/TAZ DKO.

We see the reviewer's point to provide a key functional data regarding Yap being similar to Taz (hence Taz as an example/representative). We have now provided new functional data in Fig. S4e: Forced expression of neither wt Yap nor constitutively active Yap (S127A) in *Mpp7* cKO also could not rescue the renewal fraction, similar to that by forced expression of constitutively active Taz.

The accompanying description in the main text is in ln 243-250.

3) The axis with CARM1 is proposed as a main mediator of the end point muscle phenotype. If so, could the authors use the MUSC differentiation assay to probe whether CARM1 GOF also rescue, at least to some extent, YAP/TAZ KO MUSCS? This would be important not to validate/invalidate the model, but rather to gauge its overall strength and importance in the context of the wider biology.

We have performed this requested experiment and included the data in Fig.S4f. *Carm1* overexpression in *Yap/Taz* dKO MuSCs partially rescues the progenitor fraction, but not the renewal fraction. We described this result and suggested an explanation in the result section (ln 247-252) and elaborated further in the discussion (ln 376-385).

Dear Dr. Fan,

Thank you for the submission of your revised manuscript to our editorial offices. I have now received the report from the referee that was asked to re-evaluate the study, you will find below. As you will see, referee #2 now fully supports the publication of the study in EMBO reports. Going through your p-b-p-response, I also consider the points of referee #1 as adequately addressed.

Before I can proceed with formal acceptance, I have these editorial requests I ask you to address in a final revised manuscript:

- Please shorten the abstract to not more than 175 words.

- Please provide individual production quality figure files as .eps, .tif, .jpg (one file per figure), of the main figures. Please upload these as separate, individual files upon re-submission, without their legends. Please add the legends to the main manuscript text file (for the order see below).

- We updated our journal's competing interests policy in January 2022 and request authors to consider both actual and perceived competing interests. Please review the policy <https://www.embopress.org/competing-interests> and update your competing interests if necessary. Please name this section 'Disclosure and Competing Interests Statement' and put it after the Acknowledgements section.

- We now use CRediT to specify the contributions of each author in the journal submission system. CRediT replaces the author contribution section. Please use the free text box to provide more detailed descriptions and do NOT provide your final manuscript text file with an author contributions section. See also our guide to authors: <https://www.embopress.org/page/journal/14693178/authorguide#authorshipguidelines>

- Please add up to 5 keywords and order the manuscript sections like this, using these names:

Title page - Abstract - Keywords - Introduction - Results - Discussion - Methods - Data availability section - Acknowledgements - Disclosure and Competing Interests Statement - References - Figure legends

- Please use our reference format:

Please note that DOIs should only be used for preprints and datasets that have not been published yet.

- Please make sure that the number "n" for how many independent experiments were performed, their nature (biological versus technical replicates), the bars and error bars (e.g. SEM, SD) and the test used to calculate p-values is indicated in the respective figure legends. Please also check that all the p-values are explained in the legend, and that these fit to those shown in the figure. Please provide statistical testing where applicable. Please avoid the phrase 'independent experiment', but clearly state if these were biological or technical replicates. Please also indicate (e.g. with n.s.) if testing was performed, but the differences are not significant. In case n=2, please show the data as separate datapoints without error bars and statistics. See also: <http://www.embopress.org/page/journal/14693178/authorguide#statisticalanalysis>

If n<5, please show single datapoints for diagrams. Moreover:

- Please note that the scale bar information in the legend of figure 6b, d is mislabeled as figure 6a, e in the manuscript. This needs to be rectified.

- Please note that the legend for figure 5e-f is mislabeled as figure 5g-h in the manuscript. This needs to be rectified.

- Please note that the statistical test information in figure 6b-e is mislabeled as 6a, c, e, f. This needs to be rectified.

- Please note that the exact p values are not provided in the legends of figures 1e-j; 2c, f-i, k; 3g-i; 4a, c-i; 5a-b, e-f; 6b-e; 7b, e-f; 8a-g.

- Please indicate the statistical test used for data analysis in the legends of figures 4e; 6b-d.

- Please note that in figures 5a-b, e-f; there is a mismatch between the annotated p values in the figure legend and the annotated p values in the figure file that should be corrected.

- Please add to each legend (main and EV figures, where applicable) a 'Data Information' section explaining the statistics used or providing information regarding replicates and scales. See:

- Please add scale bars of similar style and thickness to all microscopic images (main, EV and Appendix figures), using clearly visible black or white bars (depending on the background). Please place these in the lower right corner of the images themselves. Please do not write on or near the bars in the image but define the size in the respective figure legend. Presently, some scale bars are too thin or outside of the images. Please improve.

- Please provide one pdf file with Supplementary data named 'Appendix'. The Appendix should have page numbers and needs to include a table of content on the first page (with page numbers) and legends for all content. Please follow the nomenclature Appendix Figure Sx, Appendix Table Sx etc. throughout the text (for the callouts), and also label the figures and tables in the Appendix file according to this nomenclature.
- Supplementary Table S1 is a s dataset. Please upload this as dataset file (named 'Dataset EV1') with its legend on the first TAB of the excel file. Please also update the call-out for the table accordingly.
- There is a Table S2 mentioned several times in the methods section, but there is no related file uploaded. Please add this (depending on the content) either to the Appendix, the reagents and tools table, or upload it as a dataset (see above; this would then be Dataset EV2). Please also update the callouts respectively.
- All Materials and Methods need to be described in the main text using our 'Structured Methods' format, which is required for all research articles. According to this format, the Materials and Methods section should include a Reagents and Tools Table (listing key reagents, experimental models, software, and relevant equipment and including their sources and relevant identifiers), uploaded as separate file, followed by a Methods section in which we encourage the authors to describe their methods using a step-by-step protocol format with bullet points, to facilitate the adoption of the methodologies across labs. More information on how to adhere to this format as well as downloadable templates (.doc) for the Reagents and Tools Table can be found in our author guidelines (section 'Structured Methods'):

- Please make sure that all figure panels (main and Appendix figures) are called out separately and sequentially.
- It seems that Figure 1C, or parts of it, is re-used in Figure 2E, Figure 6B in Appendix Figure S6E and Appendix Figure S4G in Appendix Figure S5E. Please make sure this is correctly indicated in the respective figure legends. Presently, the reuse 1C/2E is not properly indicated, whereas the re-use in Appendix Figure S6E is wrongly attributed to 6C, and the re-use in Appendix Figure S5E is wrongly attributed to Appendix Figure S4F. Please check.
- Thank you for providing the source data. However, there are these discrepancies (compared to the source data checklist): 5D and 5E are in the checklist, while 5E and 5F are provided; 6E is in the checklist while 6F is provided. Please check.

In addition, I would need from you uploaded separately:

Best,

Referee #2:

I think the authors made a reasonable effort to answer my questions, and the data are now more complete for any reader to gauge on their interpretation.

All editorial and formatting issues were resolved by the authors.

Chen-Ming Fan
Carnegie Institution for Science
United States

Dear Dr. Fan,

I am very pleased to accept your manuscript for publication in the next available issue of EMBO reports. Thank you for your contribution to our journal.

Yours sincerely,
